# Radionuclide tracing based in situ corrosion and mass transport monitoring of 316L stainless steel in a molten salt closed loop

Yafei Wang [1,2] ✉, Aeli P. Olson[3], Cody Falconer[4,5], Brian Kelleher[5], Ivan Mitchell[5], Hongliang Zhang [4], Kumar Sridharan[1,4], Jonathan W. Engle [3] & Adrien Couet [1,4]

In the study, we report an in situ corrosion and mass transport monitoring method developed using a radionuclide tracing technique for the corrosion study of 316L stainless steel (316L SS) in a NaCl−MgCl$_2$ eutectic molten salt natural circulation loop. This method involves cyclotron irradiation of a small tube section with 16 MeV protons, later welds at the hot leg of the molten salt flow loop, generating radionuclides $^{51}$Cr, $^{52}$Mn, and $^{56}$Co at the salt−alloy interface. By measuring the activity variations of these radionuclides at different sections along the loop, both the in situ monitoring of the corrosion attack depth of 316L SS and corrosion product transport and its precipitation in flowing NaCl−MgCl$_2$ molten salt are achieved. While 316L SS is the focus of this study, the technique reported herein can be extended to other structural materials being used in a wide range of industrial applications.

Corrosion and degradation of structural materials in contact with high-temperature molten salts continue to be a limiting factor for developing sustainable energy systems including Generation IV molten salt reactors[1], thermal energy storage, and concentrated solar power plants[2,3]. Typically, high-temperature materials are self-passivating and therefore protected from corrosion induced degradation by forming passive oxide layers[4]. However, these oxide layers have a strong tendency to dissolve in high-temperature molten salts, especially molten chloride and fluoride salts[5]. The absence of protective oxides results in the electrochemical dissolution of thermodynamically susceptible alloying elements into the salt[5]. This leads to the thinning of structural components and to contamination of the salt itself. Additionally, the corrosion products can plate-out on the relatively cooler sections due to the change of chemical equilibrium constants as a function of temperature, resulting in fouling and restricted flow at heat exchangers[6,7]. It is of critical importance to systematically understand the corrosion mechanism in molten salt environments via in situ monitoring the corrosion process and

degradation of materials, ensuring the materials' compatibility with molten salts.

In recent years, significant efforts have been made to understand the corrosion mechanisms of different iron- and nickel-based alloys in high-temperature molten salts in static conditions[8–12]. While static conditions are useful to down-select alloys based on their corrosion performances in well-controlled environments, further investigations must be performed in non-isothermal flow conditions. Indeed, non-isothermal flow conditions induce thermal gradient corrosion as well as erosion if flow velocities are sufficiently high[13,14], both of which represent additional degradation mechanisms in molten salt environments. Structural material corrosion testing in flowing conditions dates back to the Aircraft Reactor Experiment, followed by the Molten Salt Reactor Experiment at Oak Ridge National Laboratory (ORNL) in the 1950s and 1960s[15–17] where corrosion studies were mainly carried out in closed natural circulation loops[13,14,18,19]. In many of these studies, samples were suspended in various sections of molten salt loops[20–22] followed by post-corrosion examination of the samples. Small-scale

[1]Department of Engineering Physics, University of Wisconsin-Madison, Madison, WI 53706, USA. [2]School of Nuclear Science and Engineering, Shanghai Jiao Tong University, Shanghai 200240, China. [3]Departments of Medical Physics and Radiology, University of Wisconsin-Madison, Madison, WI 53705, USA. [4]Department of Materials Science and Engineering, University of Wisconsin-Madison, Madison, WI 53706, USA. [5]TerraPower, LLC, Bellevue, WA 98008, USA. ✉e-mail: itsme@sjtu.edu.cn

natural circulation loops, or microloops, were recently developed by TerraPower, where the loop pipe itself served as the corrosion testing sample[23]. In all of the studies reported to date, corrosion, and mass transport mechanism investigations are based on the post-test sample characterizations, such as microstructural examination and/or sample weight change measurements. Using these methods, the corrosion rates of different materials and precipitations of corrosion products can be evaluated at given exposure time. However, these approaches are not conducive to understanding the dynamic corrosion processes in flow conditions where elucidating corrosion kinetics under multiple driving forces requires in situ techniques. Thus, it is crucial to develop an effective method to monitor in situ the corrosion and mass transport in a molten salt loop to unveil the dynamic corrosion mechanisms at play.

This gap in knowledge motivates the development of an experimental approach capable of in situ monitoring the corrosion and mass transfer in molten salt loop. Electrochemistry technique and laser spectroscopy such as laser-induced breakdown spectroscopy could be used for the in situ corrosion monitoring in molten salt[24–26] through the measurement of the concentrations of dissolved corrosion products in salts. However, the concentration distributions of corrosion products at different locations of the loop are affected by flow[20]. Thus, the corrosion rate at different section of the loop cannot be simply represented by the corrosion product concentrations at different locations of the loop. In situ gamma spectrometric measurements of radionuclides have been proposed to be utilized in the primary system of nuclear reactors to characterize the migration and deposition of radioactive elements during the reactor operation[27,28].

Inspired by this approach, in this work, we investigate the corrosion of 316L stainless steel (316L SS) and the mass transport of corrosion products in situ as a function of exposure time in a NaCl–MgCl$_2$ eutectic molten salt natural circulation loop using a radionuclide tracing technique. This technique involves irradiating a thinned tube section later welded into the hot leg of the molten salt loop with a 16 MeV proton beam to produce radionuclides of interest including $^{51}$Cr ($t_{1/2}$ = 27.7 d, $\gamma$ = 320 keV), $^{52}$Mn ($t_{1/2}$ = 5.6 d, $\gamma$ = 744, 936, and 1434 keV), and $^{56}$Co ($t_{1/2}$ = 77.2 d, $\gamma$ = 846, 1037, and 1238 keV) along the tube thickness. These radionuclides emit characteristic gamma rays

upon decay which are detected, allowing location tracking as functions of exposure time. Thus, the material corrosion and elemental mass transport can be assessed in situ by monitoring the activity variations of the radionuclides at the irradiated tube location and/or at other parts of the loop. This in situ corrosion monitoring system is coupled with a molten salt microloop design[23]. By using the developed radionuclide tracing method, the in situ monitoring of the corrosion attack depth of 316L SS at the hot leg and corrosion product transport in the microloop has been achieved. This study represents a successful achievement to monitor the corrosion of materials in situ in molten salt flowing conditions and sheds new light on the corrosion mechanism of materials in non-isothermal flowing conditions.

## Results

### Molten salt microloop

As an alloy for high-temperature system, 316L SS was selected for the construction of the molten salt microloop. The tubing material of the loop served as the corrosion testing specimens and no extra testing coupons were introduced into the loop. The loop was naturally circulated with the hot leg maintained at 620 °C while the coldest section stabilized at around 500 °C during operation. About 150 g of NaCl–MgCl$_2$ eutectic salt (58.5 mol% NaCl–41.5 mol% MgCl$_2$, melting point: 445 °C) provided by ORNL was loaded inside the loop. Driven by the temperature gradient, the loop flowed from the hot leg to the cold leg as indicated in Fig. 1a. As mentioned in the "Methods" section, the Transient Simulation Framework of Reconfigurable Models (TRANS-FORM) code, developed at ORNL[29], was utilized in this study to model the natural circulation of the microloop. Using the power to heaters as inputs while the measured temperatures by multiple thermocouples around the loop as outputs, the molten salt flow rate was determined to be about 6.3 cm/s.

### Radionuclide generations and distributions

To produce the radionuclides, a small tube section (about 1.5 cm in length) at the mid-section of the hot leg of the microloop was thinned down and irradiated with a 16 MeV proton beam (Fig. 1b). This small tube section was previously thinned down to about 150 µm as shown schematically in Fig. 1c. Figure 1d displays the thickness of the thinned

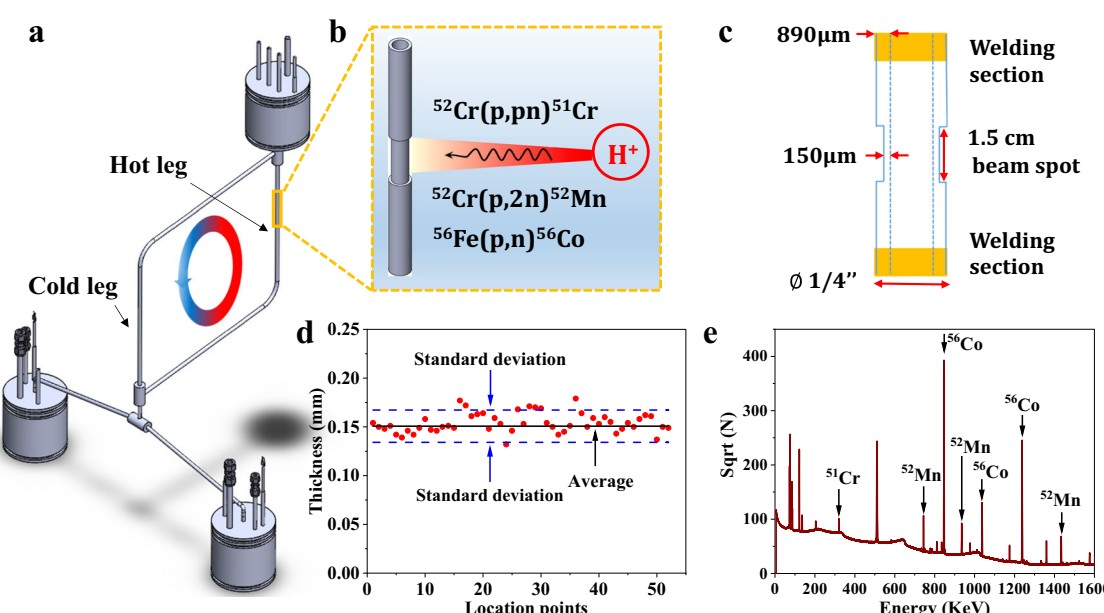

**Fig. 1 | Radionuclide tracing system coupled with the molten salt microloop.**
**a** Schematic illustration of the 316L SS natural circulation microloop.
**b** Radionuclides generation by proton beam. **c** Schematic of the thinned down tube section. **d** Thickness of the thinned down tube section. **e** Gamma spectrum obtained on the irradiated tube section by HPGe detector. Source data are provided as a Source Data file.

section measured at different locations. As illustrated in the figure, the tube was thinned down relatively uniformly over the entire thinned section length. It is expected that the radionuclides $^{51}$Cr, $^{52}$Mn, and $^{56}$Co will be produced along the tube thickness. These radionuclides emit characteristic gamma rays through their decay process, which were detected in the post-irradiated tube section and the salt flowing through other sections of the loop using two High Purity Germanium (HPGe) detectors. Figure 1e shows a representative gamma-ray spectrum obtained from the irradiated tube section after irradiation, where the peaks of $^{51}$Cr, $^{52}$Mn, and $^{56}$Co are clearly displayed.

The reaction rates for the formation of radionuclides vary as a function of the degrading energy of the proton beam, resulting in varying concentration of radionuclides along the tube thickness after irradiation. Based on the cross-sections of radionuclides production reaction as a function of proton beam energy[30], the radionuclide activity profile was determined as a function of the position in the sample using Eq. (1)[31]

$$A = N \cdot x \cdot \sigma \cdot I \cdot (1 - e^{-\lambda t}) \tag{1}$$

where $A$ is the activity in Bq, $N$ represents the number of target atoms per cm$^3$, $x$ is the distance traveled by the incident proton beam in cm, $I$ is the total number of particles incident on the target in protons/s, $\sigma$ is the reaction cross-section of interest in cm$^2$ which was determined from semi-empirical predictions of the TALYS code[30], $\lambda$ is the associated decay constant in s$^{-1}$, and $t$ is the time of exposure in seconds. As shown in the inset of Fig. 2a, the outer surface of the irradiated tube section was defined as the origin and the inner surface as the thickness of 150 μm. The modeled activity profiles in Bq/μAh are plotted in Fig. 2a as a function of location in the irradiated 316L SS alloy sample. The results show that the activity of $^{52}$Mn and $^{56}$Co increase slightly with the tube thickness. On the contrary, the activity of $^{51}$Cr sharply decreases with the thickness. There is a sudden change on the activity profiles of $^{52}$Mn and $^{51}$Cr at about 120 μm, which is due to the significant change in $^{52}$Mn and $^{51}$Cr production cross-sections at 14 MeV (energy of 16 MeV proton beam is attenuated to about

14 MeV when passing through the 316L SS at the thickness of 120 μm, see Supplementary Fig. 1).

Radionuclides may diffuse in the alloy due to the high operation temperature and elemental corrosion at the salt/alloy interface, altering the activity profiles (Fig. 2a) described above during the in situ corrosion study. Based on the initial activity profiles as shown in Fig. 2a, the evolution of radionuclides can be modeled over time. Taking $^{52}$Mn as an example, a simple diffusion model based on Fick's law[32] was utilized. By using the boundary conditions of no-flux at the tube outer surface and of zero Mn concentration at the salt/alloy interface, and assuming the diffusion coefficient of Mn in 316L SS to be $10^{-19}$ m$^2$/s[33], the concentration profile (equivalent to the activity profile) of $^{52}$Mn after 260 h (i.e., the loop operation time) was modeled and shown in Fig. 2b. It was found that $^{52}$Mn barely evolved at the hot leg, with about 99.7% of the $^{52}$Mn activity still present after 260 h. Consequently, the activity profile change by thermally driven diffusion or diffusion-induced corrosion into the salt is likely negligible. Thus, the activity loss after decay correction observed experimentally should be mainly induced by surface recession rather than by diffusion-induced corrosion. To assess surface recession, the "relative activity" was defined as the ratio of the activity remaining in the tube after a certain thickness of the tube had recessed, assuming no thermally driven diffusion of radionuclides as verified above, to the initial activity of each radionuclide in the irradiated tube, mathematically as a function of thickness $x$:

$$A_r(x) = \frac{\int_{op}^{x} A(x) dx}{\int_{op}^{ip} A(x) dx} \tag{2}$$

where $op$ is the outer surface position of the tube and $ip$ is the inner surface position of the irradiated tube.

Based on the calculated activity profile as displayed in Fig. 2a, the relative activity profiles of $^{51}$Cr, $^{52}$Mn, and $^{56}$Co as a function of the recessed layer thickness were derived as shown in Fig. 2d. The concentration (also activity) profile of $^{52}$Mn shown in Fig. 2b displays the

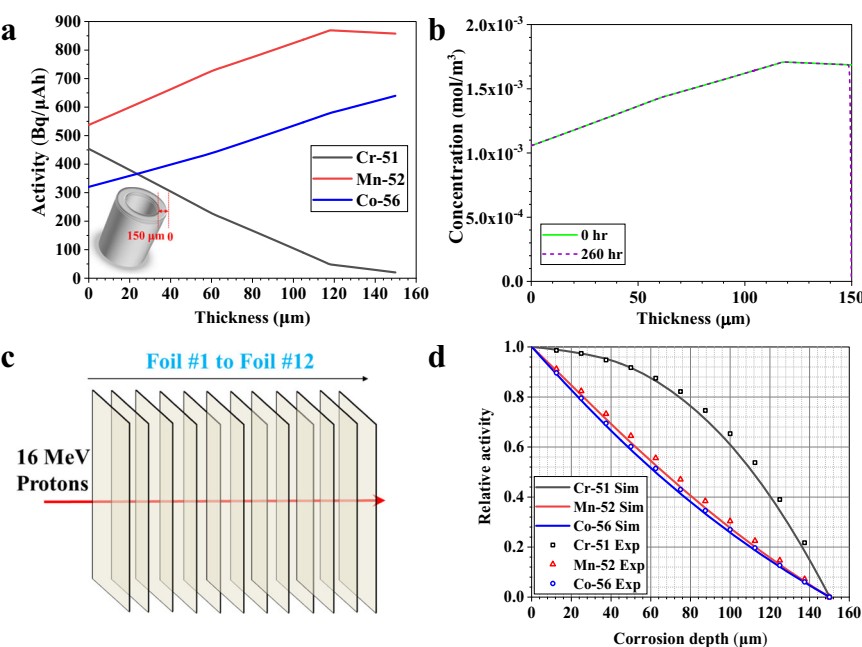

**Fig. 2 | Acquisition of activity and relative activity profiles of radionuclides.** **a** Activity of radionuclides along the irradiated tube thickness, the inset is the schematic of the irradiated tube section. **b** Concentration profiles of $^{52}$Mn along the irradiated tube thickness before and after loop operation due to the elemental diffusion. **c** Schematic of the irradiation of a stack of 12 slices of 316L SS foils. **d** Relative activity profile of radionuclides at different recession depths. Source data are provided as a Source Data file.

negligible diffusion-induced $^{52}$Mn loss. Therefore, the relative activity profile in Fig. 2d would not be affected by the diffusion brought by high temperature and elemental corrosion at the salt/alloy interface during the loop operation, and can be utilized to determine the corrosion attack (i.e., surface recession) depth. Thus, one can theoretically measure the radionuclide activity remaining in the tube during exposure, calculate the relative activity, and derive the tube recession rate in situ. However, this framework relies on the exact relative activity profiles of these radionuclides. To validate these profiles, 12 slices of 316L SS foils with the thickness of 12.5 µm each were stacked together (overall thickness: 150 µm) and irradiated with 16 MeV protons as shown in Fig. 2c. The activity of each 316L SS foil was measured by a HPGe detector. Removing one one slice of foil each time from foil #1 to foil #12 is an equivalent process of 12.5 µm of tube thickness being corroded or recessed. Through dividing the activity of the remaining foils after each removal by the total activity of the 12 slices of foils, the relative activity at the recessed depths of 12.5, 25, 37.5, 50, 62.5, 75, 87.5, 100, 112.5, 125, 137.5, and 150 µm were calculated, respectively. Figure 2d shows the comparison of the experimental relative activity data points obtained after the foil irradiation experiment with the modeled relative activity profile obtained by Eq. (1). Both results agree quite well, lending confidence that the relative activity measurements are a reliable indicator of corrosion induced surface recession rate.

## In situ corrosion monitoring

The molten salt microloop circulated naturally for about 260 h with the hot leg maintained at 620 °C and the coldest section at around 500 °C until it succumbed to corrosion product buildup and clogging. Subsequently, the temperature in the cold leg decreased to values below the NaCl–MgCl$_2$ melting point, resulting in the loss of natural circulation. The operation time of 260 h is sufficient to activate the thermal gradient driven corrosion in the loop based on the material characterizations of different tube sections along the loop: deposition of corrosion products at cold leg, severe corrosion attack at hot leg, and negligible corrosion at the medium temperature sections (top and bottom sections of the loop) as observed in Supplementary Fig. 2 and discussed in detailed later. The gamma-ray spectra of the irradiated tube were acquired continuously with a same time interval during the natural circulation process using an Ametek Ortec ICS-P4 HPGe detector. By analyzing the full energy peaks of $^{51}$Cr, $^{52}$Mn, and $^{56}$Co in the obtained gamma-ray spectra, the activity of these three radionuclides was derived as a function of time. Considering the existence of natural decay process of each radionuclide, the measured activity was decay-corrected to the end of bombardment (EOB) when each radionuclide was generated (see Supplementary Fig. 3). Through dividing the initial activity (decay-corrected to EOB) at the start of corrosion experiment, the relative activity of each radionuclide as a function of time was determined and displayed in Fig. 3. The decay-corrected initial activity of $^{52}$Mn, $^{51}$Cr, and $^{56}$Co are 2863.8 ± 7.0 kBq (limit of detection (LOD): 5.6 kBq, $\gamma$ = 320 keV), 288.6 ± 5.9 kBq (LOD: 29.2 kBq, $\gamma$ = 936 keV), and 2756.5 ± 9.6 kBq (LOD: 38.5 kBq, $\gamma$ = 1037 keV), respectively (data uncertainty is only sourced from counting statistical calculation of the full energy peak area of gamma-ray spectra).

Figure 3a shows the relative activity variation of $^{52}$Mn as a function of exposure time measured from the irradiated tube section at the hot leg of the loop. It is observed that the relative activity decreases slightly as a function of exposure time, indicating a loss of $^{52}$Mn from the tube due to molten salt corrosion. The activity loss of about 70 kBq (see Supplementary Fig. 3) at the irradiated tube section is close to the

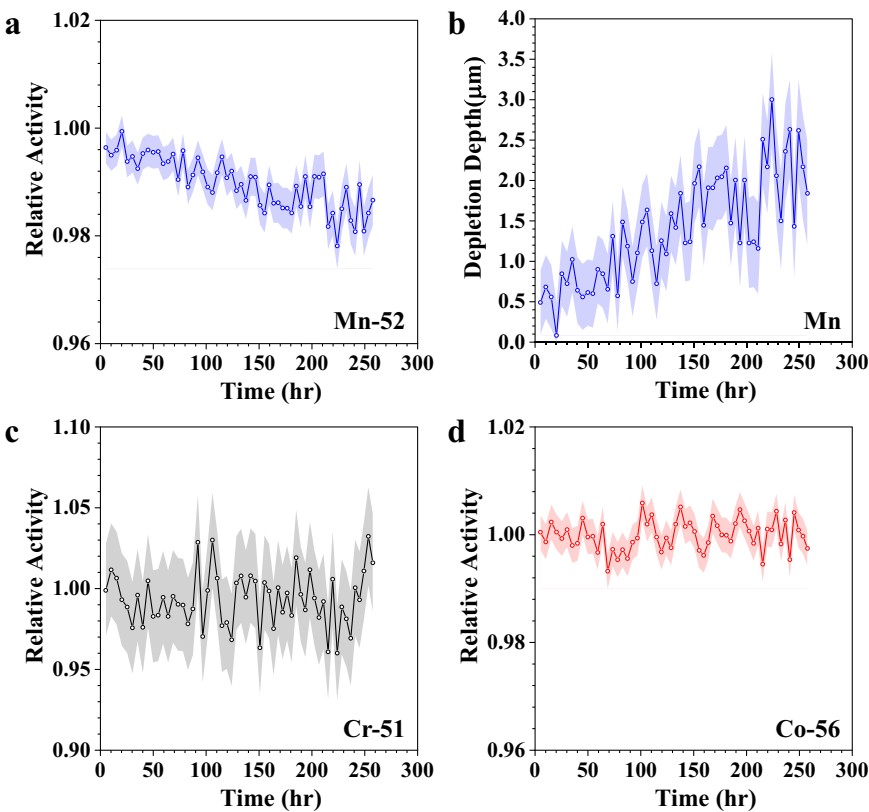

**Fig. 3 | In situ corrosion monitoring of 316L SS in the molten salt loop by radionuclide tracing. a** Relative activity variation of $^{52}$Mn with time during the loop operation process. **b** Derived recession depth variations of Mn in 316L SS with time during the loop operation process. Relative activity variations of $^{51}$Cr (**c**) and $^{56}$Co (**d**) with time during the loop operation process. The shaded area is the data uncertainty sourced from counting statistical calculation as defined by Eq. (9) in the "Methods" section. Source data are provided as a Source Data file.

detected total activity (about 68 kBq) of $^{52}$Mn in the post-corrosion salt. The relative activity of $^{52}$Mn decreases down to about 98% at the end of the loop operation. It means the activity loss of $^{52}$Mn in the hot leg is about 2%. This is about an order of magnitude higher than the modeled diffusion-induced activity loss as discussed above, indicating the corrosion of Mn mainly results from surface recession. Using Fig. 2d, the recession depth of Mn in 316L SS was extracted based on the determined relative activity and presented in Fig. 3b, and it is found to be about 3.5 μm at maximum at the end of loop operation.

The relative activity variations of $^{51}$Cr and $^{56}$Co as a function of exposure time in the irradiated tube section at the hot leg are shown in Fig. 3c, d, but no obvious activity loss is observed. Mn is thermodynamically more susceptible to corrosion in molten chloride salt than Cr since the standard Gibbs free energy of formation of MnCl$_2$/Mn ($G_{\mathrm{MnCl_2/Mn}} = -363$ kJ/mol) is lower than CrCl$_2$/Cr ($G_{\mathrm{CrCl_2/Cr}} = -277$ kJ/mol)[34,35]. This could be one factor contributing to the stable relative activity variation of $^{51}$Cr as a function of exposure time. However, due to the low composition of Mn in 316L SS (about 2 wt%), Cr typically will also dissolve into molten salt during the molten salt corrosion in the form of chromium divalent or trivalent ions as reported in previous studies[36–38], and later observed in Fig. 4. Therefore, the lack of Cr activity variation as a function of exposure time should be mainly attributed to the much lower concentration of the $^{51}$Cr, relative to $^{52}$Mn, at the inner diameter of the irradiated tube (see Fig. 2a). Consequently, any $^{51}$Cr activity loss induced by $^{51}$Cr dissolution into the salt during the loop operation is beyond the sensitivity of the gamma-ray detector. The relative activity of $^{56}$Co also remains relatively constant during exposure. This is expected since the standard Gibbs free energy of formation of CoCl$_2$/Co ($G_{\mathrm{CoCl_2/Co}} = -238$ kJ/mol) is much higher than the other main constituent elements of 316L SS such as Cr ($G_{\mathrm{CrCl_2/Cr}} = -277$ kJ/mol) and Fe ($G_{\mathrm{FeCl_2/Fe}} = -279$ kJ/mol) based on the thermodynamic database collected by HSC Chemistry 6.0[34,35]. As a result, Co was less likely to corrode during the operation of the molten salt loop.

Post-corrosion material characterization was performed on different parts of the loop after corrosion testing. Figure 4a displays the scanning electron microscope (SEM)/energy-dispersive X-ray spectroscopy (EDS) imaging of the cross-section of the post-corroded tube from the hot leg, close to the irradiated section. A severe corrosion attack was observed at the salt/alloy interface with a surface morphology typically observed in flowing molten salt corrosion[17]. Slight dissolutions of Mn and Cr, and to a lesser extent Fe were observed, while Ni is relatively enriched at the interface. In addition, based on the SEM/EDS analyses taken at different locations, it appears that the corrosion occurring at the hot leg is relatively heterogeneous, as already observed in the previous study[23]. The surface of the alloy does not recess homogeneously, likely because of heterogeneous dissolution of thermodynamically susceptible elements. Consequently, the surface morphology evolves and the pores, also called wormholes[39], are observed. In addition to the recessed layer, there are also regions exhibiting negligible corrosion. As shown in Fig. 4a, the depth at which pores are observed can be as high as 10 μm in some areas (see the top of the SEM image in Fig. 4a), while corrosion at other areas was not clearly observed (see the bottom of the SEM image in Fig. 4a). Scanning transmission electron microscope (STEM) characterization was further performed on the intense corrosion attack zone and results are shown in Fig. 4b. The results confirm that the corrosion proceeded via the preferential leaching of the thermodynamically unstable elements. Early studies[40,41] proposed that salt can infiltrate into the alloy subsurface regions and further corrode the alloy by dissolving the electrochemical susceptible elements. This is evidenced by the STEM–EDS point scans in this study showing that the Cr and Mn concentrations in the remnants of the alloy were significantly reduced as displayed in Fig. 4c.

With the radionuclide tracing method, another ICS-P4 HPGe detector was utilized to measure the activity variations of radionuclides at three different locations: P$_1$, P$_2$, and P$_3$ around the loop during its operation (Fig. 5a). These measurements areas were unshielded loop tube sections of about 5 cm in length. This system allowed for the characterization of the transport and possible redeposition of the corrosion products. In this study, the thickness of the lead sheet shield around the loop is 6.4 mm while that of the lead tube set on the HPGe detector is 3 cm. Based on the half value layer (the thickness of the material at which the intensity of the gamma-ray is reduced by 50%) of lead as a function of gamma-ray energy reported in the literature[42], the half value layer of lead for $^{51}$Cr ($\gamma = 320$ keV), $^{52}$Mn ($\gamma = 936$ keV), and $^{56}$Co ($\gamma = 1037$ keV) are about 0.18, 0.8, and 0.92 cm, respectively. The thickness of the lead sheet around the main radioactive source in the loop (the irradiated tube section), 6.4 mm, is about 3.6 times that of the half value layer of lead for $^{51}$Cr. This means only

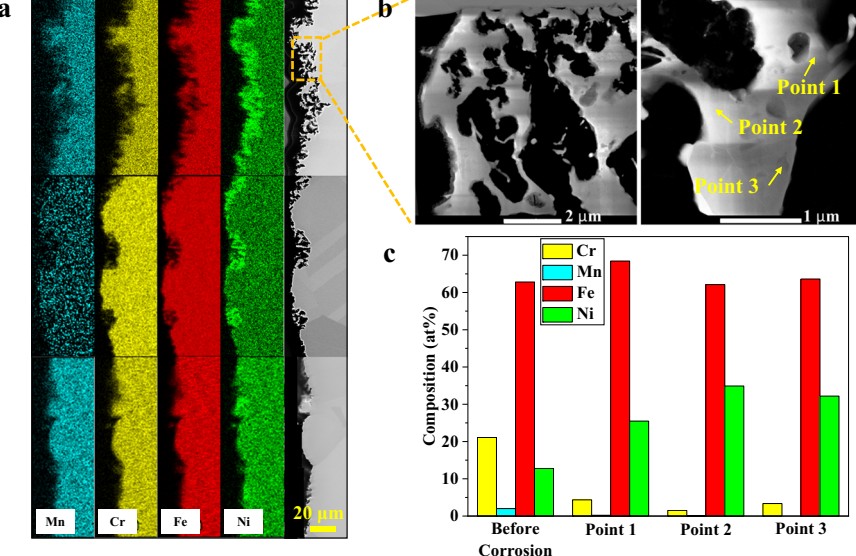

**Fig. 4 | Characterizations of samples after flow loop corrosion testing. a** SEM image of the tube cross-section close to the irradiated section at the hot leg of the loop after loop operation. **b** STEM HAADF images of corroded layer. **c** STEM–EDS point scans of the main elements' compositions in the remnants of the 316L SS alloy. Source data are provided as a Source Data file.

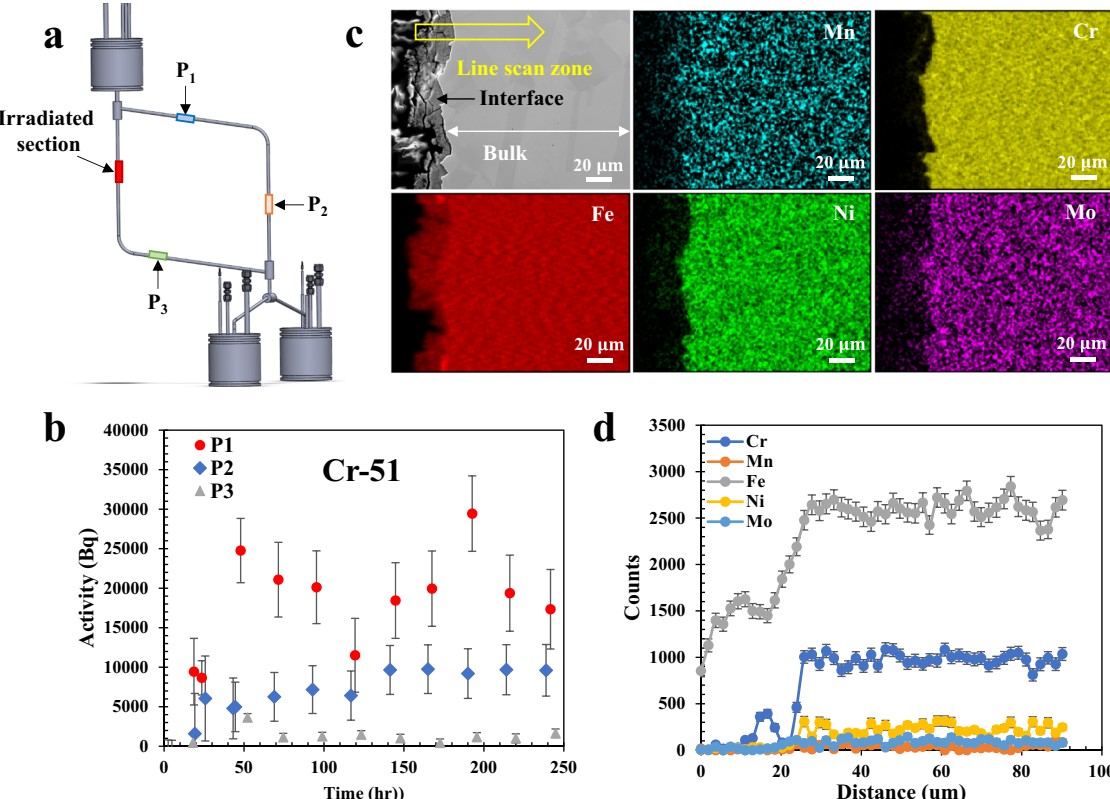

**Fig. 5 | Transport and deposition of corrosion products along the molten salt loop. a** Detecting points of the HPGe detector along the loop (highlighted area are focused for data acquisition). **b** Activity variations of ⁵¹Cr at three different locations along the loop during its operation (error bar is the data uncertainty sourced from counting statistical calculation defined by Eq. (9) in the "Methods" section). **c** SEM/EDS imaging on the cross-section of the tube section from cold leg. **d** EDS line scan on the marked zone shown in (**c**) (error bar is the data uncertainty originating from instrumental measurement sensitivity). Source data are provided as a Source Data file.

about 8% of the ⁵¹Cr gamma rays are able to escape the irradiated tube lead shielding. The much thicker lead tube set on the HPGe detector serves as an additional collimator to only measure gamma rays originating from the loop sections directly in front of the detector. In addition, the distance between the detector (for activity measurement at P1, P2, and P3) and the irradiated tube section can decrease the interference of gamma rays emitting from the irradiated tube section as well. Therefore, the shielding design in this study is sufficient to shield gamma rays originating from ⁵¹Cr and the measured activity of ⁵¹Cr for P1, P2, and P3 should be barely interfered by other loop sections. For ⁵²Mn and ⁵⁶Co, about 57% and 62% of the original gamma rays would transmit through the lead sheet, respectively. Although the thickness of the lead tube collimator (3 cm) set on the detector can shield 93% of the gamma rays from ⁵²Mn, and 90% of the gamma rays from ⁵⁶Co, a non-negligible gamma-ray intensity from the irradiated tube would be measured at P1, P2, and P3. Considering the low quantities of the dissolved ⁵²Mn and ⁵⁶Co into the salt, the measured activity of ⁵²Mn and ⁵⁶Co at P1, P2, and P3 would be heavily interfered by the irradiated tube section. Indeed, there are no obvious activity variations observed for these two radionuclides as a function of exposure time at P1, P2, and P3 (see Supplementary Fig. 4).

Figure 5b shows the activity variations of ⁵¹Cr as a function of exposure time at these three different locations. The activity levels of ⁵¹Cr at these three locations are decreasing from P₁ to P₃. This result is likely due to the decrease in radionuclide concentration within the salt, resulting from the deposition of corrosion products along the loop. While the ⁵¹Cr activity loss was not detectable in the hot leg, ⁵¹Cr activity was detected in the salt. However, the level of the detected ⁵¹Cr activity in the salt is much lower than that in the irradiated tube. It is worth noting here that the ⁵¹Cr activity in the irradiated tube is one

order of magnitude lower than ⁵²Mn and ⁵⁶Co, and that the measured ⁵¹Cr activity in the salt at P1, P2, and P3 is on the order of the random activity fluctuations of ⁵¹Cr at the irradiated tube section (about 20 kBq, see Supplementary Fig. 3). This is likely the reason why the activity loss of ⁵¹Cr was not statistically detected in the tube, i.e., the activity loss is within the detection uncertainty in the irradiated tube activity. Another interesting phenomenon is the lack of ⁵¹Cr activity in the salt (and on the tube) at P₃, right before entering the hot leg. This means that all the Cr dissolved from the hot leg redeposited along the loop within the same cycle. Basically, there is no recirculation of activated corrosion products in the loop. This is consistent with findings of a previous study that the overall corrosion in the hot section equals to the precipitation in the cold section, over an entire closed loop[43]. This is also evidenced by the increased activity of ⁵¹Cr at P₂ as a function of exposure time, which results from the deposition of Cr at that location, in addition to the activity from the flowing salt. To further verify the depositions of corrosion products at the cold leg, SEM/EDS analysis was performed on the cross-section of a tube section from the cold leg after corrosion testing and the results are shown in Fig. 5c. A deposited layer is clearly visible at the alloy/salt interface. EDS mapping and line scan (Fig. 5d) illustrate that the deposited layer is rich in Fe, with little Cr. The Cr deposition at the cold leg observed by post-test material characterization is qualitatively consistent with the in situ result obtained by radionuclide tracing.

## Discussion
The corrosion attack of the 316L SS alloy in this study can be schematically divided into three zones as displayed in Fig. 6: (I) the potentially dissolved layer (i.e., surface recession) which cannot be detected by typical material characterization methods such as SEM/

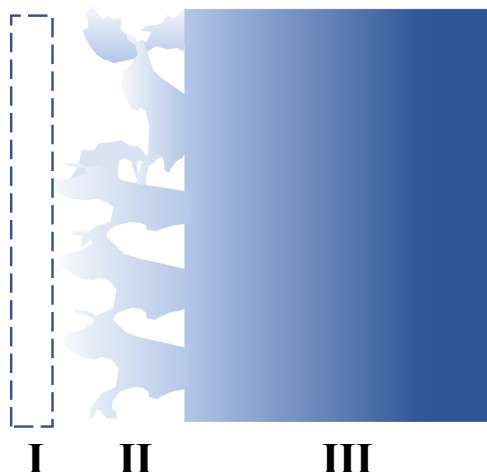

**I    II                III**

**I: Potentially dissolved layer**
**II: Heterogeneous recession layer**
**III: Bulk**

**Fig. 6 | Schematic of the corrosion attack of alloy in molten salts.** Zone I is the potentially dissolved layer of alloy, Zone II is the heterogeneous recession layer of alloy, and Zone III is the alloy bulk.

EDS/TEM; (II) the heterogenous surface recession layer with bulk material partially dissolved; and (III) bulk material. Elemental composition gradient might exist in Zones (II) and (III) because of diffusion as displayed in Fig. 6. However, based on the results from the in situ detection as discussed above and the post-corrosion material characterization by SEM/EDS as shown in Fig. 4a, there are no significant corrosion induced diffusion profiles within the corroded layer. Therefore, the recession depth given by $^{52}$Mn activity variation should be the overall tube recession depth, i.e., the recession depth of the sample itself. It is worth noting that the radionuclide tracing method developed in this study measures the overall corrosion rather than localized corrosion as observed by post-corrosion characterization. Therefore, the surface recession depth of 316L SS by flowing NaCl–MgCl$_2$ eutectic molten salt measured by this method is an overall evaluation of the corrosion behavior of the entire irradiated tube section at the hot leg. The surface recession depth determined from activity loss is averaged over the entire probed tube surface although it is clear from post-corrosion microscopy characterizations that the attack is relatively heterogeneous with a complex surface morphology development. The in situ radionuclide detection method accounts for the fully dissolved layer (Zone I in Fig. 6), the heterogenous surface recession layer (Zone II in Fig. 6), and the barely attacked substrate. Consequently, while the surface recession depth obtained by the radionuclide tracing method is about 3.5 μm, the corrosion attack can be as deep as 10 μm locally (as displayed in Fig. 4a).

The corresponding 316L SS weight loss from a surface recession depth of 3.5 μm is about 2.8 mg/cm$^2$. This is about three to four times higher than the 316L SS weight loss of about 0.8 mg/cm$^2$ observed in static NaCl–MgCl$_2$ eutectic salt at 600 °C for 240 h[44]. However, the salt volume of about 150 g used in our natural circulation loop is also about three times of the salt volume (60 g) in the static corrosion experiment[44]. In addition, considering the influence of flow on corrosion and the slight difference of the experimental conditions of temperature and corrosion testing time, these two corrosion results are relatively comparable. This is an indirect comparison between the corrosion of static and flow conditions, a direct comparison will be needed in the future to study the corrosion behaviors under static and flow by setting same experimental conditions. A corrosion study of

316L SS in NaCl–MgCl$_2$ molten salt natural circulation loop has been similarly performed for 1000 h[23], but the 316L SS corrosion performance was only evaluated by post material characterization methods (i.e., at the end of loop operation). It is worth noting that the corrosion attack depth of 316L SS at the same location in the loop after 1000 h of natural circulation was found to be about 20 μm, which is in qualitative agreement with the results reported in this study for 260 h of exposure, validating the corrosion results reported in this study.

The radionuclide tracing method developed in this study essentially is a technique to measure the variation of activity occurring by material's loss. Cr and Mn are two main constituent elements for high-temperature (>700 °C) alloys and are susceptible to dissolution in molten salts. Therefore, $^{51}$Cr and $^{52}$Mn could be used for the in situ corrosion and mass transport monitoring of high-temperature alloys when exposed to molten salt environments. $^{56}$Co is also produced via nuclear reactions such as $^{56}$Fe(p,n)$^{56}$Co, although Co is not an element of interest to measure materials' corrosion, since it is relatively thermodynamically stable and should experience little dissolution in molten salts. However, the produced $^{56}$Co can serve as a reference to monitor the stability of the detector during corrosion testing because of its thermodynamical stability and relatively longer half-life. The original distribution of the generated radionuclide tracer in the material is also crucial for the in situ corrosion monitoring. Unfortunately, in our study of 316L SS, the concentration of $^{51}$Cr at the tube inner diameter is too low and the corresponding activity loss was not detectable. A higher concentration of radionuclide tracers distributed at the salt/alloy interface, would make the radionuclide tracing method more sensitive to in situ corrosion monitoring of $^{51}$Cr. On the other hand, a few micrometers of Mn depletion can be detected through the activity measurement of $^{52}$Mn because of the relatively high concentration of $^{52}$Mn at exposed surface. However, due to the high energy of the gamma-ray originating from $^{52}$Mn, its signal at other positions in the loop were heavily interfered by the irradiated tube section at the hot leg. Consequently, the mass transport study of Mn in the loop was not feasible. As a surrogate, the detectable $^{51}$Cr in the loop can be used to study the mass transport of corrosion products although it is not feasible for in situ measurement of depletion depth. The complementary of the activity measurements of multiple radionuclides could be an ideal way for the in situ corrosion evaluation and mass transport monitoring of corrosion products in molten salts.

This study was devoted to the development of a radionuclide tracing based in situ corrosion monitoring method for high-temperature molten salt system and demonstrated its unique capabilities. Indeed, surface recession was measured in situ for the first time in a high-temperature molten salt recirculation loop system. It was also shown experimentally for the first time that all Cr dissolving from the hot leg redeposits on the cold leg in recirculation. The novelty of this study is not only the application of the tracer idea in an untapped field (molten salt corrosion), but the development of this approach itself which consists in purposefully created radioactive corrosion tracers in materials to study dissolution and mass transport in recirculation loops. This approach could be applied to many other corrosion systems such as pressurized water and liquid metal coolants. It should be noted that the radionuclides produced in this study are from proton irradiation. Neutron irradiation in reactor could create a more homogeneous radionuclide profile through the tube thickness, and this would be an interesting path moving forward to provide more insights to future corrosion studies.

## Methods
### Salt preparations
The NaCl–MgCl$_2$ eutectic salt used in this study was prepared with 58.5 mol% anhydrous NaCl salt and 41.5 mol% anhydrous MgCl$_2$ salt. The salt was homogenously mixed and purified by ORNL. The trace impurities of the NaCl–MgCl$_2$ eutectic salt mixture were identified by

ICP-MS analysis in which the most prominent impurity elements and the corresponded concentrations were 0.79 ppm Li, 5.44 ppm S, 15.7 ppm K, 6.99 ppm Ca, 0.03 ppm Cr, 0.01 ppm Mn, 0.13 ppm Fe, 0.23 ppm Ni (weight). The oxygen level in the salt is around 300 to 400 ppm (weight).

## Molten salt microloop design and operation

The molten salt microloop in this study was built with 316L SS based on the design by TerraPower[23]. The nominal chemical composition of 316L SS is given in Supplementary Table 1. While the schematic of the microloop and its auxiliary equipment is shown in Supplementary Fig. 5. The loop itself is the corrosion testing material to be investigated. The dimensions of the whole loop frame are 32″ × 32″ × 32″ and the loop body is a 9.9″ × 12.3″ parallelogram. The tube diameter of the loop is 0.25″ with a wall thickness of 0.035″. The loop is micro, requiring only two, 96 cm³ batches of salt for continuous operation. The molten salt flow in the microloop was established through natural circulation by setting up a hot leg and a cold leg sections, as detailed in ref. 23. In this design, a temperature gradient across the harp-shaped portion of the microloop generates salt density gradients resulting in buoyant forces that drive the natural circulation of the molten salt. There are three tanks in the microloop as shown in Supplementary Fig. 5. The bottom of the microloop has a flush salt tank and a primary salt tank. The flush salt was stored in the flush salt tank to "clean" the residual weld oxides and debris from the microloop construction before the test, while the primary salt was stored in the primary tank for the actual test. The surge tank at the top is designed for safety to allow for excess molten salt volume to expand, if necessary, without pressure buildup. The microloop's main body was wrapped with nichrome heating wire to heat up different sections of the loop to their target temperatures and initiate natural circulation. Thermocouples were welded onto the tube outer wall surface at different sections of the loop to monitor the temperature. The two legs at the bottom of the microloop (see Supplementary Fig. 5) were equipped with ceramic heaters serving as freeze valves. To fill molten salt into the loop, the salt in the primary salt tank (or flush salt tank) was first melted and then primed into the loop's main body by argon gas pressure through a gas line connected to each tank. The power of the ceramic heaters serving as the freeze valves was switched to a temperature lower than the salt melting point immediately after the molten salt is primed into the loop. Following this, the molten salt in that section froze, blocking the tube, such that the molten salt in the loop would not flow back into the tank. Once the main body of the loop was filled with molten salt, natural circulation was initiated by setting a temperature gradient along the loop using nichrome heating wire and monitoring the thermocouples at different sections of the loop. After the natural circulation was initiated, the loop was operated continuously with about 5 psig argon cover gas to avoid the possible uptake of atmospheric air into the loop through microleaks.

## Radionuclide production

To enable radionuclide tracing, a small section of tube must be irradiated, activating the tube's inner diameter contacting the salt. Thus, the proton beam used to generate radionuclide had to be of sufficient energy and intensity at the inner tube location to overcome the activation cross-section energy thresholds and obtain relatively large reaction rates. However, the 16 MeV proton beam energy available for this experiment was too low to generate a useful activation profile for radionuclides of interest across the entire tube thickness (about 890 μm). A thickness of about 150 μm would generate relatively significant radionuclide concentration at the inner surface of the tube to enable gamma-ray detection. A tube section about 15 cm in length was cut from the hot leg of the microloop. The central part of that section (about 1.5 cm in length, corresponding to

beam spot size for irradiation) was thinned down to about 150 μm using a lathe. The thickness of the thinned tube section was measured at different locations by a Magna-Mike® 8600 magnetic probe. These locations include both the points being parallel to the 1.5 cm long section and along the lateral tube surface. To irradiate the thinned 316L SS section, a custom target holder with slit collimator was designed and fabricated (see Supplementary Fig. 6). Before irradiation, the tube was placed into the target holder and sealed at the two ends by Swagelok fittings. By adjusting the position of the tube, the thinned 1.5 cm section of the tube can be aligned below the slit of the collimator. The slit is 0.8 mm wide, which is well within the 9.5 mm width of the cyclotron proton beam collimators. The collimators are about 6 cm upstream of the slit, and no external optical focusing elements act on the beam after extraction by stripping foil. As a result, the collimator width has no effect on the beam profile incident on the tube. The target holder was connected to the cyclotron beam port by a KF flange (see Supplementary Fig. 6). The two Swagelok fitted ends were connected to a water line outside the cyclotron, allowing internal water cooling to the tube and stop the ion flux. The tube was irradiated with a 16 MeV proton beam using a General Electric PETtrace cyclotron located at the Wisconsin Institutes for Medical Research. The beam intensity was initially 2 μA, but was raised after 2 h of initial irradiation to 3 μA for another 6 h, resulting in a beam fluence of 20 μAh.

The displacement per atom (dpa) induced by the beam is quite small, about 0.04 dpa, based on the Stopping and Range of Ions in Matter (SRIM) analysis[45]. In 316L SS, a proton irradiation dose of 1–1.5 dpa is considered close to the saturation of the dislocation loops and the radiation induced segregations phenomenon, where effects on post-corrosion rate can be observed[46]. In this study, the damage is about 25 times less than that. In addition, irradiation-induced point defects quickly anneal out (recombine or diffuse to sinks) at 620 °C during the corrosion exposure. Consequently, that low level of radiation in the material is not expected to affect post-irradiation corrosion rate in this study.

The radionuclides of interest as tracers for in situ corrosion monitoring in the microloop are ⁵¹Cr, ⁵²Mn, and ⁵⁶Co. ⁵¹Cr was produced via the nuclear reactions shown in Eq. (3)

$$^{52}\text{Cr}(p,pn)^{51}\text{Cr}$$

$$^{52}\text{Cr}(p,2n)^{51}\text{Mn} \rightarrow (\text{decay}) \rightarrow {}^{51}\text{Cr}$$

$$^{54}\text{Fe}(p,\alpha)^{51}\text{Mn} \rightarrow (\text{decay}) \rightarrow {}^{51}\text{Cr} \tag{3}$$

while ⁵²Mn was produced through the reaction (4).

$$^{52}\text{Cr}(p,n)^{52}\text{Mn} \tag{4}$$

⁵⁶Co was produced through the reactions shown in Eq. (5)

$$\begin{aligned} ^{56}\text{Fe}(p,n)^{56}\text{Co} \\ ^{57}\text{Fe}(p,2n)^{56}\text{Co} \end{aligned} \tag{5}$$

Although there are a few characteristic gamma emissions with the generated radionuclides ⁵²Mn and ⁵⁶Co, the gamma emissions at 936 keV for ⁵²Mn while 1037 keV for ⁵⁶Co were selected for their quantifications to avoid the interference from the nearby signals.

## In situ corrosion and mass transport monitoring system

The 15 cm 316L SS tube containing the thinned and irradiated section of around 1.5 cm in length at its middle was welded back to the microloop using an orbital welding technique. Before welding, the potential influence of heat affected/fusion zone induced by

welding on the corrosion and irradiation responses of the tube was characterized. This was achieved by performing orbital welding on prototypical 316L SS tube samples (see Supplementary Fig. 7). The welded tube section was cut axially, mounted in epoxy, polished using SiC paper with different grit size from 320 to 1200 grits, etched with aqua regia, and then analyzed by optical microscopy (see Supplementary Fig. 7). Results showed that the total length of fusion zone was about 2.5 cm, while the heat affected zone was not detectable at this magnification. To avoid the potential effects of the weld-induced microstructure on the corrosion of the proton activated section, the total section of the tube used was 15 cm, meaning the proton activated section was about 6.75 cm away from the weld section and would not be affected.

A pressure test of the microloop was performed to confirm the absence of leaks after welding the irradiated tube section back to the hot leg of the loop. Then, the nichrome heating wire and insulation material were rewrapped back on the tube section. The salt loading of the microloop was performed inside an inert glovebox filled with argon gas ($O_2 < 2$ ppm, $H_2O < 2$ ppm) since $NaCl–MgCl_2$ salt are hygroscopic and very sensitive to moisture and oxygen in the air. Once the salt was loaded into the tank, the microloop was sealed by Swagelok fitting caps and pressurized with about 5 psig argon gas. The loop was moved out from the glovebox while the pressurized argon gas aids in preventing uptake of atmospheric air and moisture through potential microleaks.

The natural circulation of the microloop was initialized by setting up a temperature gradient. The time between the EOB and the start of loop operation is about 264 h to allow the short half-life radionuclides to decay out. Once the natural circulation of the molten salt was established, the molten salt started to corrode the inner surface of the tube and the thermodynamically unstable elements (including radionuclides) started dissolving into molten salt. The radionuclides produced in the irradiated tube as a part of the loop would decay by emitting gamma rays. The in situ corrosion monitoring of the loop was achieved through measuring the activity variations of different radionuclides as functions of time and location by two Ametek Ortec ICS-P4 HPGe detectors, which are p-type aluminum windowed ORTEC GEM detectors with a relative efficiency at 1333 keV of about 10% and a measured gamma peak full width at half maximum (FWHM) at 1333 keV of 1.99 keV. The HPGe detector was calibrated for energy and efficiency using a europium-152 check source of known activity. One HPGe detector faced the irradiated tube section at the hot leg of the microloop to monitor tube corrosion and the other HPGe detector was used to measure the activity variations of radionuclides at different locations around the loop (see Supplementary Fig. 8), allowing the characterization of the transport and redeposition of the corrosion products. To mitigate the interference of gamma rays from other locations/directions on the HPGe detectors, lead shields were installed in the system (see Supplementary Fig. 8). The thickness of the lead sheet shield around the loop is 6.4 mm while that of the lead tube set on the HPGe detector is 3 cm. The gamma spectra for the cold leg of the loop were measured before the corrosion experiment to verify the ability of the lead shielding to absorb gamma rays. As shown in Supplementary Fig. 9, no $^{51}$Cr full energy peak ($\gamma = 320$ keV) was found, indicating the lead shielding is sufficient to avoid the interference of $^{51}$Cr from the irradiated tube section. However, strong signals of $^{52}$Mn and $^{56}$Co resulted from the interference of the irradiated tube section were detected. Therefore, it may not be feasible to study the transports and redepositions of $^{52}$Mn and $^{56}$Co in this study.

### Gamma-ray spectra data analysis

To get the activity of $^{51}$Cr, $^{52}$Mn, and $^{56}$Co from the obtained gamma-ray spectra, the total counts (or area) of the full energy peaks for each radionuclide were calculated. In this study, the peak channel ±3 FWHM

was selected as the region of interest (ROI)[47]. The background area $B$ of the full energy peak is given by Eq. (6):

$$B = \left( \sum_{i=l}^{l+(n-1)} C_i + \sum_{i=h-(n-1)}^{h} C_i \right) \frac{h-l+1}{2n} \tag{6}$$

where $l$ is the ROI low limit, $h$ is the ROI high limit, $C_i$ is the counts of channel $i$, $n$ is the number of background points which is set to 5 in this study as suggested in the literature[47]. The adjusted gross area $A_{ag}$ is the sum counts of all the channels in the ROI but not used in the background according to Eq. (7)[47]

$$A_{ag} = \sum_{i=l+n}^{h-n} C_i \tag{7}$$

The net area $A_n$ is the adjusted gross area minus the adjusted calculated background, as follows[47]

$$A_n = A_{ag} - \frac{B(h-l-(2n-1))}{(h-l+1)} \tag{8}$$

The uncertainty of the net peak area, $\sigma_{A_n}$, is given by[47]:

$$\sigma_{A_n} = \sqrt{A_{ag} + B\left(\frac{h-l-(2n-1)}{2n}\right)\left(\frac{h-l-(2n-1)}{h-l+1}\right)} \tag{9}$$

The activity of each radionuclide was derived through dividing the net area $A_n$ by the calibrated detector efficiency and branching fraction. The associated uncertainty of activity was simply converted from $\sigma_{A_n}$ using error propagation rule. It should be noted that the variations of temperature or humidity of the room or detector high voltage (HV) might also contribute to uncertainty[48]. However, considering the stable basement environment where the experiment was performed (see room temperature variation shown in Supplementary Fig. 10) and the fixed HV value set through the whole experiment, the uncertainty contributions from temperature, humidity, and detector HV were neglected in this study.

### Material characterization

SEM coupled with EDS was performed at the Wisconsin Center for Nanoscale Technologies to characterize the samples after corrosion experiment. Zeiss LEO 1530 device equipped with an energy-dispersive X-ray spectrometer and Pathfinder software was used for the SEM/EDS analysis. Samples for the STEM analysis were prepared using a standard lift-out technique by an FEI Helios PFIB G4 FIB/FESEM Focused Ion Beam (FIB) instrument in the Materials Science Center at the University of Wisconsin-Madison. The high-angle annular dark-field scanning transmission electron microscopy (HAADF-STEM) images and EDS in an FEI Titan G2 80–200 (S)TEM equipped with the EDS detector. The microscope was operated at an acceleration voltage of 200 kV, with a probe current of 300 pA and a probe convergence angle of 21 mrad. A dwell time of 30 s was used for each point scan to achieve good statistics. Simultaneous HAADF and EDS acquisition were performed using the Bruker Esprit software. The cross-section of each tube to be characterized was mounted with epoxy, ground with SiC abrasive papers of different grit sizes up to 1200 grit, and then polished on the polishing pads by 3 μm, 1 μm diamond suspensions, and 0.04 μm colloidal silica suspension.

### Loop circulation rate modeling

TRANSFORM code developed at ORNL[29] was utilized in this study to model the natural circulation of the microloop. In the simulation (see Supplementary Fig. 11), each pipe was modeled as a single connected loop, cold leg section was modeled as pipe with a single heat transfer

surface on which a circulation boundary condition was imposed, the heaters in the loop were modeled with internal heat generation equivalent to the output data obtained from the loop operation process. Through adjusting the molten salt flow rate in the PID controller added in the TRANSFORM model to maintain the temperature at different locations of the loop to best match the measured thermocouple data in the loop operation (see Supplementary Fig. 12), the molten salt flow rate during the loop operation was determined. More detailed information regarding the simulation process can be found in refs. 22,29.

## Data availability
The data that support the findings of this study are included in the main text and supplementary information files. Source data are provided with this paper.

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

## Acknowledgements

This study is funded by the US Department of Energy NEUP Award Number DE-NE0008904 (to A.C.). The authors gratefully acknowledge use of facilities and instrumentation supported by NSF through the University of Wisconsin Materials Research Science and Engineering Center (DMR-2309000). Y.W. would like to thank the financial support from the National Natural Science Foundation of China (Grant No. 12305395 to Y.W.) and the startup research funding from Shanghai Jiao Tong University with the Grant No. WH220402024 (to Y.W.). A.P.O. gratefully acknowledges support from the USA NIH T32 NRSA Institutional Predoctoral Training Fellowship 2T32CA009206-41 (to A.P.O.) and NIH F31 Ruth L. Kirschstein Predoctoral Individual National Research Service Award F31CA239617 (to A.P.O.). The content is solely the responsibility of the authors and does not necessarily represent the official views of the NIH. The authors also wish to thank Dr. Jinsuo Zhang from Virginia Tech and Zhi Qin from Institute of Modern Physics, Chinese Academy of Sciences for the valuable discussions, ORNL for providing the TRANSFORM code, and Nicholas Crnkovich from University of Wisconsin-Madison for performing the SRIM analysis.

## Author contributions

Y.W. and C.F. constructed the experimental facility with the help from B.K. and I.M.; Y.W. and C.F. performed corrosion experiments, data analysis, and sample preparations; Y.W. and H.Z. performed the material characterization; Y.W. and A.P.O. conducted activity measurements of radionuclides; Y.W. and A.C. performed the thermodynamic modeling work and drafted the manuscript. A.C., K.S., and J.W.E. conceived of the original project and oversaw its execution, providing regular guidance.

## Competing interests

The authors declare no competing interests.
