## [Peer Review File · Nature Communications]

Radionuclide Tracing Based in situ Corrosion and Mass Transport Monitoring of 316L Stainless Steel in a Molten Salt Closed LoopREVIEWER COMMENTS

Reviewer #1 (Remarks to the Author):

This paper presents a novel and potentially useful new method for monitoring corrosion of metals by molten salts in situ. It is strong enough to be published with some minor revisions. My comments/suggestions are listed below.

Line 34: What is a "string tendency"?

Some quantification of the initial purity of the NaCl-MgCl₂ is needed with respect to water or OH⁻.

I don't think it is necessary to write Fick's Law in one dimension (equation 2). You can make reference to it in the text without writing out the equation. It is very well known, and there is nothing unique about your version of it.

line 206-207: I would suggest that the reason Mn activity decreases and Cr activity does not is that MnCl₂ is thermodynamically more stable than CrCl₂. The standard reduction potential of Mn is lower than Cr, indicating Mn will oxidize before Cr.

I'm puzzled as to why you made measurements at only the cold leg and transition from cold to hot and hot to cold. Why not a measurement in the hot leg? I would be interested to see if there is a clear evidence of corrosion in the hot leg and deposition in the cold leg as reported recently by Raiman et. al.

At what position in the loop is the data in Figure 3 collected? I kind of figured out that it must be where the irradiation was done, but this is not clearly stated. It should be pointed out in Figure 5(a).

The first paragraph of the Discussion section reads like the conclusions. I would delete or move this content to the conclusions.

Reviewer #2 (Remarks to the Author):

The research presented in the article is mostly devoted to the description and application of a new unique technique for studying the corrosion of materials in liquid melts, which allows in situ determination of corrosion rates as well as the further transfer of corrosion products in a loop with the melt. Also of interest are the experimental data obtained on the corrosion of 316L Stainless Steel in a NaCl-MgCl₂ eutectic molten salt, which actually demonstrate the capabilities of the proposed technique.

The work is very relevant in the field of studying the corrosion of materials in melts, and opens up new experimental approaches to such research. The work is original, the research context is justified by relevant references to previous works.

The research methodology is described in great detail and can be reproduced (with reservations for the unique modes of some stages: for example, parameters of neutron irradiation of a sample pipe (target), dimensions of the pipe (target), etc.).

A complex of microstructural studies, as well as a transparent and clear analysis of the data obtained in the experiment, fully justify the methodological approach of the authors. The work meets the high scientific and technical standard in the field of corrosion experiments.

Separately, I would like to note a very detailed description of the experimental features of the research and a qualitative, well-founded analysis of various methodological factors of the research that influence the results of the experiment (for example, correct consideration of the possible influence of corrosion in the welding area of the target pipe sample). Overall, the work makes an excellent impression and has no comments on my part.

Reviewer #3 (Remarks to the Author):

This paper presents interesting work on in-situ corrosion monitoring in a molten salt loop using the monitoring of radioactive corrosion products activated by the proton beams, followed by the surface analysis of corroded metals. However, the study has several critical issues that need to be addressed.

1. Most importantly, it is unclear what new information might be suggested or revealed by this radionuclide tracing method. Concentrations cannot be directly interpreted due to differences in the half-lives and gamma-ray energies of the radionuclides. It would be beneficial to determine whether these radionuclides can act as effective radioactive markers for understanding corrosion behaviors, such as dissolution and deposition, across different temperature zones of the loop. This appears to be a primary aim of the study, yet it seems to be incompletely addressed.
2. The activity measurements of activated products provide the only rough indicators of corrosion product concentrations. Counts or activity levels should be converted into concentrations to obtain more detailed corrosion-related information, achievable through careful calibration or Monte Carlo simulation.
3. It would enhance the paper if the authors briefly explained the potential advantages of radionuclide tracing methods over those of electrochemistry, laser spectroscopy, or other monitoring techniques used for molten salt conditions.
4. In Figure 2, is it valid for the activities of Mn-52 and Cr-51 to change sharply near $120\mu\text{m}$?
5. Is there a reason for the differing trends in activity according to the thicknesses of Mn/Co and Cr as shown in Figure 2(a)? Figure 2(d) suggests that the activity trends of Cr, Mn, and Co are similar when considering them in relation to the thickness.
6. Please provide the activity levels for each radionuclide at the start of the corrosion experiments and the detection limits for these activities.
7. In Figure 5(a), could you mark the measurement areas of P1, P2, and P3? In Figure 5(b), could you display the activities of Mn-52 and Co-56 at P1, P2, and P3?
8. In Figure 5(d), is there a specific reason why Cr deposition occurred only near the surface of the pipe?
9. Is it reasonable to predict the recession depths of other nuclides using the recession depth of Mn? How was the weight loss determined if the recession depths were not the same?

Reviewer #4 (Remarks to the Author):

Dear authors,

I had pleasure reading your interesting work reporting on the in-situ corrosion and mass transport monitoring in a molten salt loop. You had made quite some effort irradiating a tube section with

protons to generate activation products and the salt-metal interface and performed repeated gamma spectroscopy measurements during the course of a 260 h natural circulation experiment. While the in-situ gamma measurement technology itself is not novel (see, for example 10.1016/j.anucene.2008.12.030 or 10.1109/ANIMMA.2015.7465291) there is indeed no reported work that applied such in-situ technique for corrosion studies in molten salt systems. This is the novelty of the paper and therefore, deserves publication.

However, you seem to overestimate the abilities of gamma spectrometry and have made several conclusion that lack a scientific basis. Therefore, i recommend you to revise your manuscript based on my comments below:

Line 28: In-situ gamma spectrometry for corrosion and activation product studies is not novel. Please include a couple of references for the reader

Line 34: typo, should likely be "strong tendency", not "string tendency"

Line 36: it should read "... electrochemical dissolution of ... elements"; the word dissolution needs to be singular, since it is the dissolution of 316L containing several elements that dissolve

Line 88: This sentence is difficult to understand. The authors try to emphase that the tubing material itself served as source for specimens to evaluate corrosion behavior of 316L SS. Please rephrase accordingly

Line 92: Counterclockwise flow – this is misleading, since Figure 5 suggests otherwise. It depends on the position of the observer, so a reference point needs to be defined. Please either avoid such a description of the flow direction or clearly indicate a reference point for the observer
The introduction is well written, thank you!

Line 100 downwards:

First of all, there are a couple of details given in this section that rather should be implemented into the experimental methods section, since some of the information given does not belong to results. For example, the energy and efficiency calibration of the HPGe is typically reported under "experimental". It is recommended that the authors chose what information to present in the result section to keep a concise and clear structure.

Second, there is some information missing in this and the methods section, that would be helpful to include for the reader to understand the details of the proton irradiation and the subsequent gamma spectroscopic measurements, namely:

- for the irradiated tube section, a length of 1.5 cm is mentioned, but no width. This presumably depends on the collimator slit dimensions, which were not provided. Additionally, the beam spot size is mentioned to correspond to 1.5 cm – is this at 1σ FWHM? Please specify. Could it be assumed that the radionuclide activities along these 1.5 cm is distributed uniformly?

- there is only the total charge of 20 μ Ah given, but not the average beam current and irradiation time. What was the time between end of irradiation and start of experiment?

- it is stated that the radiation damage, indicated as displacement per atom (dpa), is "quite small" and can be neglected. Please provide a reasoning for this statement and an argumentation, how much, in terms of radiation damage affecting corrosion resistance, "quite small" actually is. A reader, who is not familiar with corrosion of materials nor effects of radiation damage, cannot judge if a proton irradiated 316L stainless steel sample will or will not be more prone to corrosion attack compared to non-irradiated 316L SS. Please provide some literature to make your point rock solid.

- Figure 1d – it is not clear to the reader at what positions the thickness of the thinned down tube

section was measured. The choice for the location points can be parallel to the 1.5 cm long section or along the lateral tube surface – please specify in the text.

- gamma spec needs a more detailed description. ICS-P4 is only a designation that the detector head came with an internal cooling system, i.e. was electrically cooled. There is no specific information about the Ge crystal given, such as relative efficiency, resolution etc. Please provide this information using the associated Ortec datasheets.

- It is stated that the detector was calibrated for energy and efficiency using an Am-241 reference source. That is by far insufficient to be able to quantify the activity of radionuclides mentioned in this manuscript. Please consult a person proficient in gamma spectrometry and elaborate on the methodology these detector(s) were calibrated with.

Some further comments:

Line 130: If I is the beam current, given in [μA], then equation (1) should also contain the time t before the brackets. The product $I \cdot t$ in [$\text{A s} = \text{Coulomb}$] is the total charge transmitted to a target

Line 135: The activity profiles in Figure 2a is given in $\mu\text{Ci}/\mu\text{Ah}$, not $\text{nCi}/\mu\text{Ah}$. consider using SI units Bq

Line 152: What is diffusion-induced radioisotope corrosion? Chemically and physically, there is no difference between radioactive and stable isotopes of elements in a vast majority of processes. Diffusion-induced corrosion will influence the erosion of stable Mn-55, present 2 wt% in 316L SS, the same way as it would for Mn-52. The radioactive isotopes in this manuscript are used as in-situ tracers, that behave exactly as the according element.

However, radiotracers decay – and the sentence in line 153 is somewhat misleading. The authors mention it later, but unless decay correction is applied, the activity loss observed experimentally will mainly come from decay loss (even for Co-56). Please rephrase this sentence and add decay correction.

Line 175 – Figure 2a: Why there is a sudden change in the slope for all three curves at roughly 120 μm depth? Figure 2b: Why do you plot the concentration of Mn-52, and not stable Mn-55 (see also comment above)? It is not clear to the reviewer why it was important to prove the negligible effect from diffusion induced Mn loss using an additional figure. An explanation in the text could have been sufficient. Figure 2c – what is the form of the Bragg curve passing 150 μm of 316L SS? Would it be possible to indicate the mean proton energy exiting Foil 12?

Line 183: It is stated that the experiment was running for 260 h, after which the cold leg temperature reached values below the eutectic melting point. It is a bit unclear if this was out of intention or it happened accidentally, stopping the whole endeavor prematurely. Could you indicate what was the true reason for the melt temperature to drop?

Line 192 downwards:

The authors make here conclusions on the type of the underlying Mn corrosion and the recession depth based on the observed decrease in measured, decay-corrected Mn-52 activity over time. It is not clear to the reviewer why – as common practice in scientific studies like this – an uncertainty estimation was not performed first before making any claims towards interpretation of the observed phenomena. It is stated that a decrease of 2% is observed for the Mn-52 activity between start and end of the experiment. However, gamma spectrometry typically comes with an uncertainty of at least 3-5% for quantitative activity measurements, originating from the uncertainty of the calibration source activity, reported emission probability uncertainties, the counting statistics, etc. Even in relative measurements, where detector efficiency and emission probabilities cancel out, short, medium and long-term variations of temperature or humidity of the room or detector HV induce an uncertainty contribution that cannot be neglected. Please consult H.C. Lepy et al.: Uncertainties in gamma-ray spectrometry, *Metrologia*, Vol 52(3) 2015 for further details.

It would have been important to have installed a fixed check source attached to the experimental setup that would provide "reference" lines from a long-lived isotope, i.e. Co-60 or Eu-152 for each HPGe measurement to monitor its stability over time. It is obvious from Figure 1a, that especially in the last 40 h of the experiment there is a considerable variation of the measured, decay corrected Mn-52 activity. If this is due to counting statistics, an oscillation effect in the measurement over time or due to other effects, needs to be checked in more detail. However, the paper cannot be accepted as such due to the obvious neglect of the uncertainty contribution. Please rewrite this section and the resulting conclusions accordingly.

Some further comments:

Line 203: It appears that in Figure 3c, the activity for Cr-51 seems to rise – an effect that is likely explained considering different uncertainty contributions. It is unclear why here the authors state that it "remains stable", while the apparent relative change is much more pronounced if compared to Figure 3a. It seems someone wants to trick the reader plotting the y-axis in 3a

Line 214: Reference [29] is a website that might likely disappear in the future. Please use a published book or a scientific paper, for example I. Barin: Thermochemical Data of Pure Substances, VCH, Berlin (1995).

Line 217: Figures 3a-d contain two distinct graphs each, however no explanation is given what is the difference. Avoid the usage of the unit [Ci], use the SI [Bq] instead. It is even recommended not to report activities in these graphs at all and rather refer to activity ratios relative to experiment start. What is the correlation factor between the full energy peak count rates of Mn-52 and Cr-51? There appears to be a dip in activity for both isotopes at 150 h, is this a coincidence?

Line 242 – Figure 4c: It would be beneficial to keep the color coding of the elements as in Figure 4a and plot on the x-axis point 1,2,3 and before and use the color for the elements. This should improve the readability of that data.

Line 252: There is no full clarity how interference from the irradiated pipe section or other parts of the microloop was suppressed for activity determinations at P1, P2 and P3. From Figure 4S there appears to be some shielding at various positions at various thickness. Were there tests performed if the installed Pb shielding thickness is sufficient to sufficiently suppress gamma lines originating from Cr-51, Mn-52 or Co-56? It is likely much easier to do so for Cr-51 compared to the others. Would it be possible to measure the total activity of Mn-52 in the loop (P1+P2+P3) and determine the recession depth from the ratio to its initial activity at the welded section? An ISOCS characterized detector will likely be required for this purpose.

Line 256: The authors state that the Cr-51 signal is decreasing from P1 to P3, likely due to the decrease of Cr concentration within the salt. Is there any possibility to retrieve a sample of the salt to underline this statement? Cr transport is likely due to dissolution/precipitation in the eutectic melt at different positions of the loop, but in-situ gamma spectrometry is unable to determine if the origin of the Cr-51 signal is the salt, the solid-liquid interface or the 316L itself.

Line 262: The authors state that one order of magnitude less (10%) of Cr-51 activity is within the detection noise of the HPGe. Are you sure about the absolute activities stated in Figure 5b? Anyhow, it is still confusing to read your conclusions drawn with Mn-52, where only 2% of loss was measured ...

Line 267: Reference [34] applies to LBE corrosion, where other phenomena dominate. Comparing your observations to effects reported for liquid metal corrosion is like comparing apples and pears. Please look out for a reference describing molten salt corrosion.

Line 277 – Figure 5c: For someone not familiar with SEM images, it would be helpful to indicate where is the bulk and where the interface layer.

Line 312 downwards:

The authors give references to other works that compares corrosion depth to the (statistically questionable) stated recession depth of 3.5 μm . To the reviewers opinion, however, the most important outcome of the work is the confirmation of Cr dissolution and re-deposition at different sections of the loop using this in-situ technique. It is surprising why there was no further data provided for Co-56 and Mn-52, since these could have given further insights on the corrosion behavior of these two elements in molten salt loops.

Reactor irradiation of stainless steel samples would create the possibility to introduce samples of much higher activity and more homogeneous distribution for isotopes Cr-51 and Fe-59. Although this technique is associated with a number of challenges, the authors could provide more insights about potential benefits for future molten salt corrosion studies, giving examples of existing ones from nuclear power plants (see references above).

It is a stony way towards improvement, but many thanks for your hard work!

Authors' Responses to the Review Comments

Manuscript NCOMMS-23-47066

Title: Radionuclide Tracing Based in situ Corrosion and Mass Transport Monitoring of 316L Stainless Steel in a Molten Salt Closed Loop

Authors: Yafei Wang, Aeli P. Olson, Cody Falconer, Brian Kelleher, Ivan Mitchell, Hongliang Zhang, Kumar Sridharan, Jonathan W. Engle, Adrien Couet

Nov 30, 2023

We thank the editor and the reviewers for their time, effort, and valuable comments. The manuscript has been revised based on the reviewers' comments and the revised part has been highlighted in yellow. We hope that these revisions improve the paper such that you deem it worthy of publication in Nature Communications.

Response to the reviewers' comments

Reviewer #1:

- **Summary Comment:** This paper presents a novel and potentially useful new method for monitoring corrosion of metals by molten salts in situ. It is strong enough to be published with some minor revisions. My comments/suggestions are listed below.
- ✓ **Response:** We thank the reviewer for the valuable comments and the recommendation of acceptance of our manuscript in Nature Communications. The manuscript has been revised accordingly based on the comments and suggestions given by the reviewer (see below) and are highlighted in yellow in the revised manuscript.
- **Comment 1:** Line 34: What is a "string tendency"?
- ✓ **Response:** We thank the reviewer for pointing out this typo. It should be "strong tendency" not "string tendency" and has been corrected in the revised manuscript. At the same time, the whole manuscript has been checked out thoroughly to confirm there is no typo left.
- **Comment 2:** Some quantification of the initial purity of the NaCl-MgCl₂ is needed with respect to water or OH⁻.

- ✓ **Response:** The impurity of water or OH⁻ in salt can be analyzed by measuring the contents of O or H. According to our collaborators at Oak Ridge National Laboratory who have supplied the salt for this study, the oxygen level for the chloride salts they produced is typically around 300-400 ppm (weight). However, they did not analyze H in chloride salts due to instrument limitations (and to the authors knowledge, there are no techniques accepted by the community capable of quantify hydrogen in chloride salts). For more information on the characterization methods by ORNL (here on fluoride salt but applicable to chloride salts), please see D. Sulejmanovic, J.M. Kurley, K. Robb, S. Raiman, Validating modern methods for impurity analysis in fluoride salts, J. Nucl. Mater. 553 (2021) 152972.

The oxygen level in the salt has been added in page 14 of the revised manuscript.

- **Comment 3:** I don't think it is necessary to write Fick's Law in one dimension (equation 2). You can make reference to it in the text without writing out the equation. It is very well known, and there is nothing unique about your version of it.
- ✓ **Response:** As the reviewer commented, we have deleted the equation of Fick's law in the revised manuscript, but replaced with adding a citation (Fick, Adolph. "On liquid diffusion." Journal of Membrane Science 100, no. 1 (1995): 33-38.). The manuscript has been revised accordingly in page 5.
- **Comment 4:** line 206-207: I would suggest that the reason Mn activity decreases and Cr activity does not is that MnCl₂ is thermodynamically more stable than CrCl₂. The standard reduction potential of Mn is lower than Cr, indicating Mn will oxidize before Cr.
- ✓ **Response:** We agree with the reviewer that Mn is thermodynamically more susceptible to corrosion in molten chloride salt than Cr since the standard Gibbs free energy of formation of MnCl₂/Mn ($G_{\text{MnCl}_2/\text{Mn}} = -363$ KJ/mol) is much lower than CrCl₂/Cr ($G_{\text{CrCl}_2/\text{Cr}} = -277$ KJ/mol), and thus pure Mn will oxidize before pure Cr. This could be one factor contributing to the stable activity of ⁵¹Cr as a function of exposure time and has been added into the revised manuscript. However, due to the low composition of Mn in 316L SS, Cr typically will also dissolve into molten salt during the molten salt corrosion of 316L SS as reported in previous studies [1,2], and observed in our SEM analysis shown in Figure 4. Therefore, the lack of activity variation of ⁵¹Cr isotope should mainly result from the much lower concentration of the ⁵¹Cr, relative to ⁵²Mn, at the inner diameter of the irradiated tube (as opposed to the higher concentration of Cr element relative to Mn element in 316L SS).
- **Comment 5:** I'm puzzled as to why you made measurements at only the cold leg and transition from cold to hot and hot to cold. Why not a measurement in the hot leg? I would

be interested to see if there is a clear evidence of corrosion in the hot leg and deposition in the cold leg as reported recently by Raiman et. al.

- ✓ **Response:** We apologize for the unclear statement. The activity variations of ^{52}Mn , ^{51}Cr , and ^{56}Co displayed in Figure 3 are measured from the irradiated tube section at the hot leg of the loop and showing clear evidence of corrosion occurring at the hot leg. A clearer statement has been added in page 7 in the revised manuscript. There is also clear evidence of deposition in the cold leg as observed in Figure 5 in the manuscript.
- **Comment 6:** At what position in the loop is the data in Figure 3 collected? I kind of figured out that it must be where the irradiation was done, but this is not clearly stated. It should be pointed out in Figure 5(a).
- ✓ **Response:** Similar to our previous response to comment 5, the data in Figure 3 is from the irradiated section at the hot leg of the loop. A clearer description has been added and highlighted in yellow in page 7 of the revised manuscript. It was also added in Figure 5(a).
- **Comment 7:** The first paragraph of the Discussion section reads like the conclusions. I would delete or mov this content to the conclusions.
- ✓ **Response:** We agree with the reviewer. This paragraph has been deleted in the revised manuscript.

We thank the reviewer for carefully checking our manuscript. All these discussions above have been reflected in revised manuscript. The content revised has been highlighted in yellow in the revised manuscript.

Reviewer #2:

- **Summary Comment:** The research presented in the article is mostly devoted to the description and application of a new unique technique for studying the corrosion of materials in liquid melts, which allows in situ determination of corrosion rates as well as the further transfer of corrosion products in a loop with the melt. Also of interest are the experimental data obtained on the corrosion of 316L Stainless Steel in a NaCl-MgCl₂ eutectic molten salt, which actually demonstrate the capabilities of the proposed technique.

The work is very relevant in the field of studying the corrosion of materials in melts, and opens up new experimental approaches to such research. The work is original, the research context is justified by relevant references to previous works.

The research methodology is described in great detail and can be reproduced (with reservations for the unique modes of some stages: for example, parameters of neutron irradiation of a sample pipe (target), dimensions of the pipe (target), etc.).

A complex of microstructural studies, as well as a transparent and clear analysis of the data obtained in the experiment, fully justify the methodological approach of the authors. The work meets the high scientific and technical standard in the field of corrosion experiments.

Separately, I would like to note a very detailed description of the experimental features of the research and a qualitative, well-founded analysis of various methodological factors of the research that influence the results of the experiment (for example, correct consideration of the possible influence of corrosion in the welding area of the target pipe sample).

Overall, the work makes an excellent impression and has no comments on my part.

- ✓ **Response:** We thank the reviewer for these positive summary comments and are very grateful to the reviewer for their recognition of our work.

Reviewer #3:

- **Summary Comment:** This paper presents interesting work on in-situ corrosion monitoring in a molten salt loop using the monitoring of radioactive corrosion products activated by the proton beams, followed by the surface analysis of corroded metals. However, the study has several critical issues that need to be addressed.
- ✓ **Response:** We thank the reviewer for the positive summary comments and appreciate their time and efforts to help us improve the quality of this manuscript. Based on the review comments, we have revised the manuscript accordingly (see below). We hope that these revisions could improve the paper such that the reviewer deems it worthy of publication.
- **Comment 1:** Most importantly, it is unclear what new information might be suggested or revealed by this radionuclide tracing method. Concentrations cannot be directly interpreted due to differences in the half-lives and gamma-ray energies of the radionuclides. It would be beneficial to determine whether these radionuclides can act as effective radioactive markers for understanding corrosion behaviors, such as dissolution and deposition, across different temperature zones of the loop. This appears to be a primary aim of the study, yet it seems to be incompletely addressed.
- ✓ **Response:** This study was mainly devoted to the development of a radionuclide-tracing-based *in situ* corrosion monitoring method for high temperature molten salt system and

demonstrated its capabilities. Indeed, surface recession was measured in-situ for the first time in a high-temperature molten salt recirculation loop system. We believe the novelty of the paper is not only the application of this tracer idea in an untapped field (molten salt corrosion), but the development of this approach itself which consists in purposefully created radioactive corrosion tracers in materials to study dissolution and mass transport in recirculation loops. This approach could be applied to many other corrosion systems such as pressurized water and liquid metal coolants.

The above statement has been added in page 13 in the revised manuscript to clarify the primary aim of this study.

We are not sure we understand the comment of “concentrations cannot be directly interpreted due to differences in the half-lives and gamma-ray energies of the radionuclides” mentioned by the reviewer. We would like to clarify that all the activity (or counts) reported in the manuscript have been decay-corrected to the end of bombardment (EOB) when each radionuclide was generated based on the half-life of each radionuclide. Therefore, the concentrations of the radionuclides can be directly converted from the activity (or counts) measured from gamma-ray spectra through simply dividing by the volume of the irradiated zone. In this study, we have no capabilities to report the spatially dependent concentration of each radionuclide through the tube length or thickness *in situ*. To our knowledge, no emerging or existing techniques can provide that level of accuracy, and the development of such a technique (i.e., gamma-ray 3D imaging at the sub-millimeter scale) would be well-beyond the scope of this work.

The main objective of the radionuclide tracing based *in situ* corrosion monitoring method developed in this study is to measure the activity (decay-corrected, equivalent to concentration) of each radionuclide in the irradiated section at the hot leg of the loop. Dividing by the initial activity (decay-corrected) of each radionuclide before corrosion testing, the relative changes can be detected and the corrosion attack depth can be extracted. We are not sure we understand how gamma-ray energies could affect the interpretation of the concentration of each radionuclide as the reviewer commented. Each radionuclide has their own characteristic gamma-ray full energies. The obtained activity is based on the analysis of the characteristic full energy peak for each radionuclide.

The effectiveness of each radionuclide tracer was discussed in the discussion section of this manuscript. In fact, the choice of effective radioactive tracers would vary depending on the corrosion environment under study, e.g., long-term corrosion testing, short-term corrosion testing, flow condition, static condition, etc. We are now exploring adapting this technique to other systems and we invite the reviewer to follow our future work employing the method pioneered through this study.

- **Comment 2:** The activity measurements of activated products provide the only rough indicators of corrosion product concentrations. Counts or activity levels should be converted into concentrations to obtain more detailed corrosion-related information, achievable through careful calibration or Monte Carlo simulation.
- **Response:** We agree with the reviewer that the activity measurements (and the equivalent concentration measurements) are dependent on careful calibrations of the gamma-ray detectors and could potentially be improved through simulation. However, as what we answered to comment 1, the main principle of the radionuclide tracing based *in situ* corrosion monitoring method developed in this study is to detect the **relative** changes of the activity of each radionuclide in the irradiated section at the hot leg of the loop during its operation. Therefore, the reliability (or accuracy) of the results reported in this study should not be affected by further calibration or simulation.
- **Comment 3:** It would enhance the paper if the authors briefly explained the potential advantages of radionuclide tracing methods over those of electrochemistry, laser spectroscopy, or other monitoring techniques used for molten salt conditions.
- ✓ **Response:** We thank the reviewer for this comment. Electrochemistry technique and laser spectroscopy such as laser-induced breakdown spectroscopy (LIBS) can be used for the *in situ* corrosion monitoring in molten salt [3,4,5] through the measurement on the concentrations of the dissolved corrosion products in salts. However, the concentration distributions of corrosion products at different locations of the loop are affected by flow [6]. Thus, the corrosion attack degree of different section of the loop cannot be simply represented by the corrosion product concentrations at different locations of the loop. The authors are not aware of other monitoring techniques capable of achieving in-situ local corrosion measurements in molten salt flow conditions. A clarifying statement has been added in the introduction section of the revised manuscript to enhance its quality based on the reviewer comments.
- **Comment 4:** In Figure 2, is it valid for the activities of Mn-52 and Cr-51 to change sharply near 120 μm ?
- ✓ **Response:** The energy of the proton beam passing through the 316 L SS will decrease gradually as function of thickness. At the thickness of 120 μm , the energy of the 16 MeV proton beam is attenuated to about 14 MeV. Based on the TENDL tabulated cross sections [7] as shown below, there is a strong change on the cross sections for producing ^{52}Mn and ^{51}Cr at 14 MeV. The sudden changes of the activity profiles of ^{52}Mn and ^{51}Cr at the thickness of 120 μm are due to this sharp change in the cross sections.

This explanation has been added in page 4 in the revised manuscript.

- **Comment 5:** Is there a reason for the differing trends in activity according to the thicknesses of Mn/Co and Cr as shown in Figure 2(a)? Figure 2(d) suggests that the activity trends of Cr, Mn, and Co are similar when considering them in relation to the thickness.
- ✓ **Response:** The activity profiles for different radionuclides displayed in Figure 2(a) are calculated based on equation (1) in the manuscript. The reaction cross sections for each radionuclide are different and vary with the thickness of the tube. It is why different trends in activity for different radionuclides are observed in Figure 2(a).

Figure 2(d) shows the relative activity of each radionuclide, which was simply derived from Figure 2(a). The relative activity was defined as the ratio of the activity remaining in the tube after a certain thickness of the tube would have recessed (or been removed) from

the inner surface to the initial activity of each radionuclide in the irradiated tube. The relative activity A_r as a function of thickness x is defined mathematically as:

$$A_r(x) = \frac{\int_{op}^x A(x) dx}{\int_{op}^{ip} A(x) dx}$$

where op is the outer surface position of the tube and ip is the inner surface position of the tube.

The manuscript has been revised accordingly in page 5 to make the description clearer.

- **Comment 6:** Please provide the activity levels for each radionuclide at the start of the corrosion experiments and the detection limits for these activities.
- ✓ **Response:** The activity of ^{52}Mn , ^{51}Cr , and ^{56}Co at the start of the corrosion experiment are $77.4 \pm 0.19 \mu\text{Ci}$ ($2863800 \pm 7030 \text{ Bq}$), $7.8 \pm 0.16 \mu\text{Ci}$ ($288600 \pm 5920 \text{ Bq}$), $74.5 \pm 0.26 \mu\text{Ci}$ ($2756500 \pm 9620 \text{ Bq}$), respectively. The limit of detection (LOD) for ^{52}Mn , ^{51}Cr , and ^{56}Co are calculated to be about $0.15 \mu\text{Ci}$ (5550 Bq), $0.79 \mu\text{Ci}$ (29230 Bq), and $1.04 \mu\text{Ci}$ (38480 Bq), respectively, using the LOD determination method [8]. The initial activity and LOD have been added in page 7 in the revised manuscript.
- **Comment 7:** In Figure 5(a), could you mark the measurement areas of P1, P2, and P3? In Figure 5(b), could you display the activities of Mn-52 and Co-56 at P1, P2, and P3?
- ✓ **Response:** The tube sections of the loop corresponding to lengths of about 5 cm of unshielded area are labelled P1, P2, and P3 as marked in Figure 5(a). This has been clarified in page 9 in the revised manuscript.

Due to their relatively high energies, the signals of ^{52}Mn and ^{56}Co at P1, P2, and P3 were significantly interfered by the gamma-rays originating from the irradiated tube section at the hot leg of the loop (i.e., the signal is a convolution of the irradiated area signal and the signal from the unshielded area). It is the reason their activity was not previously displayed in the original manuscript. However, we have revised to add the activity variations of ^{52}Mn and ^{56}Co at P1, P2, and P3 to the supplementary document.

Due to the compactness of our molten salt loop setup, we were not able to install a radiation shielding capable of shielding gamma-rays originating from high-energy radionuclides. In our study, the thickness of the lead sheet shielding around the loop is 6.4 mm while that of the lead tube set on the HPGe detector is 3 cm. Based on the half value layer (the thickness of the material at which the intensity of the gamma-ray is reduced by 50%) of lead as a function of gamma-ray energy as shown below [9], the half value layer of lead for ^{51}Cr (γ

= 320 keV), ^{52}Mn ($\gamma = 936$ keV), and ^{56}Co ($\gamma = 1037$ keV) are about 0.18 cm, 0.8 cm, and 0.92 cm, respectively. The thickness of the lead sheet around the main radioactive source in the loop (the irradiated tube section), 6.4 mm, is about 3.6 times the half value layer of lead for ^{51}Cr . This means only about 8% of the ^{51}Cr gamma-rays are able to escape the irradiated tube lead shielding. The much thicker lead tube set on the HPGe detector serves as an additional collimator to only measure gamma-rays originating from the loop sections directly in front of the detector. In addition, the distance between the detector (for activity measurement at P1, P2, and P3) and the irradiated tube section can also decrease the interference of gamma-rays emitting from the irradiated tube section. Therefore, the shielding design in this study is considered sufficient to shield gamma-rays originating from ^{51}Cr and the measured activity of ^{51}Cr at P1, P2 and P3 should be barely interfered by other loop sections.

Half value layer of lead as a function of gamma-ray energy [9].

Escape ratio of gamma-rays from different radionuclides under different shields.

Shielding thickness	escape ratio		
	^{51}Cr ($\gamma = 320$ keV)	^{52}Mn ($\gamma = 936$ keV)	^{56}Co ($\gamma = 1037$ keV)
6.4 mm	8%	57%	62%
3 cm	~0	7%	10%

On the other hand, for ^{52}Mn and ^{56}Co , about 57% and 62% of the original gamma rays would transmit through the lead sheet, respectively. Although the thickness of the lead tube collimator (3 cm) set on the detector can shield 93% of the gamma-rays from ^{52}Mn , and 90% of the gamma-rays from ^{56}Co , the detector can still accept gamma-rays from its front end. There is a large amount of gamma-rays being measured at P1, P2, and P3 originating from the irradiated tube. As shown in the Figure below, there are measurable

differences in activity of ^{52}Mn and ^{60}Co at P1, P2 and P3, but the measurements are likely too convoluted to be confidently interpreted. Considering the high value of the activity of ^{52}Mn and ^{60}Co at P1, P2 and P3 as shown below, the measured activity should be mainly from the irradiated tube section. This also explained the close value of the activity of ^{52}Mn and ^{60}Co as shown below (the activity of ^{52}Mn and ^{60}Co at the irradiated tube section are close to each other). In the Figure below, the difference on the activity of ^{52}Mn and ^{56}Co at different locations resulted from the different distances between the detector and the irradiated tube section and the different shielding angle of the lead tube relatively to the irradiated tube sections for these three different locations.

Figure: Data uncertainty is not visible due to the large y-axis scale. Note that the ^{52}Mn and ^{56}Co activity at all points are quite close from one another, and that they are also quite close from one another in the irradiated tube (see activity variations of these three radionuclides in the Supplementary Document), reinforcing the fact that these radioisotope activities at P1, P2, and P3 are convoluted with the activities from the irradiated tube.

The manuscript has been revised accordingly based on the above discussion in page 10 of the revised manuscript.

- **Comment 8:** In Figure 5(d), is there a specific reason why Cr deposition occurred only near the surface of the pipe?
- ✓ **Response:** We are not sure we understand the comment about the deposition only occurring near the surface of the pipe. We interpret this comment as the observation of a “peak” in the Cr EDS line scan counts in Figure 5(d) at about 15 μm from the surface. This is different from a more gradual increase in the signal, as observed for instance with Fe. This is the authors’ understanding that the peak corresponds to a local deposition picked up by the EDS line scan and that the deposition is certainly not homogeneous at the scale of the EDS line scan. The local deposition of Cr serves as a validation for the results obtained from gamma-ray spectroscopy. In fact, because of the relatively low Cr content in 316L SS and

small quantities of Cr dissolved into molten salt during exposure, the detection of Cr deposited at the cold leg by EDS is quite challenging. On the other hand, that deposition was detected by gamma-ray spectroscopy, highlighting another benefit of using gamma-ray spectroscopy. A lack of Cr detection by EDS at the cold leg tube surface was also reported in a previous molten salt loop study by Raiman et al [10].

- **Comment 9:** Is it reasonable to predict the recession depths of other nuclides using the recession depth of Mn? How was the weight loss determined if the recession depths were not the same?
- ✓ **Response:** The recession depth given by ^{52}Mn activity variation represents overall tube recession depth (i.e., recession depth of the sample itself) since very limited alloying element diffusion profile are observed experimentally (Figure 4(a)) or via modeling (Figure 2(d)). Therefore, the recession depth is element independent, and the weight loss can be directly determined based on the recession depth. This statement has been mentioned in the discussion section in page 11 of the revised manuscript.

We thank reviewer #3 again for carefully checking our manuscript. We have attempted to consider their comments and our revisions are highlighted in yellow. We hope that these revisions improve the paper such that the reviewer deems it worthy of publication.

Reviewer #4:

- **Summary Comment:** Dear authors, I had pleasure reading your interesting work reporting on the in-situ corrosion and mass transport monitoring in a molten salt loop. You had made quite some effort irradiating a tube section with protons to generate activation products and the salt-metal interface and performed repeated gamma spectroscopy measurements during the course of a 260 h natural circulation experiment. While the in-situ gamma measurement technology itself is not novel (see, for example 10.1016/j.anucene.2008.12.030 or 10.1109/ANIMMA.2015.7465291) there is indeed no reported work that applied such in-situ technique for corrosion studies in molten salt systems. This is the novelty of the paper and therefore, deserves publication. However, you seem to overestimate the abilities of gamma spectrometry and have made several conclusions that lack a scientific basis. Therefore, I recommend you to revise your manuscript based on my comments below:
- ✓ **Response:** We thank the reviewer for the recommendation of this work and their time and efforts to improve our manuscript. We completely agree with the reviewer that the use of gamma measurement technologies on-site, or in-situ, is not new and has been used to study mass transport of radioactive species. We believe the novelty of the paper is not only the use of this technology in an untapped field (molten salt corrosion), but the approach itself

which consists in purposefully created radioactive corrosion tracers in materials to study dissolution and mass transport in recirculation loops. This approach could be applied to many other corrosion systems such as pressurized water and liquid metal coolants. Based on the review comments, we have revised the manuscript accordingly (see below). We hope that these revisions improve the paper such that the reviewer deems it worthy of publication.

- **Comment 1:** Line 28: In-situ gamma spectrometry for corrosion and activation product studies is not novel. Please include a couple of references for the reader.
- ✓ **Response:** As the reviewer commented, it is the first attempt to use the in-situ gamma spectrometry for the corrosion studies in high temperature molten salt systems although gamma-ray spectroscopy has been used in other fields before, such as mass transport of radioactive species. We do agree with the reviewer that the in-situ technique itself is not novel and have deleted the word of “novel” throughout our manuscript. In addition, a few references relevant of in-situ gamma detection technique are now provided to the reader by the addition of a new sentence “In situ gamma spectrometric measurements of radionuclides have been proposed to be utilized in the primary system of nuclear reactors to understand the migration and deposition of radioactive elements during the reactor operation [11,12]. Inspired by this approach, here we investigated the corrosion of 316L SS and the mass transport of corrosion products *in situ* as a function of exposure time in a NaCl-MgCl₂ eutectic molten salt natural circulation loop using a radionuclide tracing technique.” in the revised manuscript.
- **Comment 2:** Line 34: typo, should likely be “strong tendency”, not “string tendency”.
- ✓ **Response:** We thank the reviewer for pointing this typo. The “string tendency” has been replaced with “strong tendency” in the revised manuscript.
- **Comment 3:** Line 36: it should read “... electrochemical dissolution of ... elements”; the word dissolution needs to be singular, since it is the dissolution of 316L containing several elements that dissolve.
- ✓ **Response:** Based on the reviewer’s comment, the word “dissolutions” has been replaced with “dissolution” in the revised manuscript.
- **Comment 4:** Line 88: This sentence is difficult to understand. The authors try to emphasize that the tubing material itself served as source for specimens to evaluate corrosion behavior of 316L SS. Please rephrase accordingly.

- ✓ **Response:** We apologize for this unclear statement. To make it clear, the original sentence has been rephrased as “The tubing material of the loop served as corrosion testing specimens and no extra testing coupons were introduced into the loop”.

- **Comment 5:** Line 92: Counterclockwise flow – this is misleading, since Figure 5 suggests otherwise. It depends on the position of the observer, so a reference point needs to be defined. Please either avoid such a description of the flow direction or clearly indicate a reference point for the observer. The introduction is well written, thank you!

- ✓ **Response:** We agree with the reviewer, the circle with arrow in Figure 1(a) actually indicates the loop flow direction. In the revised manuscript, it has been rephrased as “Driven by the temperature gradient, the loop flowed from the hot leg to the cold leg as indicated in Figure 1(a)” to make it clearer.

- **Comment 6:** Line 100 downwards: First of all, there are a couple of details given in this section that rather should be implemented into the experimental methods section, since some of the information given does not belong to results. For example, the energy and efficiency calibration of the HPGe is typically reported under “experimental”. It is recommended that the authors chose what information to present in the result section to keep a concise and clear structure.

- ✓ **Response:** We agree with the reviewer. The details regarding the experimental setup have been moved to the methods section in the revised manuscript.

- **Comment 7:** Second, there is some information missing in this and the methods section, that would be helpful to include for the reader to understand the details of the proton irradiation and the subsequent gamma spectroscopic measurements, namely:
 - for the irradiated tube section, a length of 1.5 cm is mentioned, but no width. This presumably depends on the collimator slit dimensions, which were not provided. Additionally, the beam spot size is mentioned to correspond to 1.5 cm – is this at 1σ FWHM? Please specify. Could it be assumed that the radionuclide activities along these 1.5 cm is distributed uniformly?

- ✓ **Response:** We agree with the reviewer that more details should be given in the methods section. The slit is 0.8 mm wide, which is well within the 9.5 mm width of the cyclotron proton beam collimators. The collimators are about 6 cm upstream of the slit, and no external optical focusing elements act on the beam after extraction by stripping foil. As a result, the collimator width has no effect on the beam profile incident on the tube. This is

why the beam spot size is not described more precisely. This description has been added into page 15 of the revised manuscript.

- **Comment 8:** there is only the total charge of 20 μAh given, but not the average beam current and irradiation time. What was the time between end of irradiation and start of experiment?
- ✓ **Response:** The beam intensity was initially 2 μA , but was raised after 2 hours of initial irradiation to 3 μA for another 6 hours, resulting in a beam fluence of 20 μAh . The time between end of irradiation and start of experiment is about 264 hours to allow the short half-life radionuclides to decay out. This has been added in page 15 and page 17 of the revised manuscript.
- **Comment 9:** It is stated that the radiation damage, indicated as displacement per atom (dpa), is “quite small” and can be neglected. Please provide a reasoning for this statement and an argumentation, how much, in terms of radiation damage affecting corrosion resistance, “quite small” actually is. A reader, who is not familiar with corrosion of materials nor effects of radiation damage, cannot judge if a proton irradiated 316L stainless steel sample will or will not be more prone to corrosion attack compared to non-irradiated 316L SS. Please provide some literature to make your point rock solid.
- ✓ **Response:** Following the reviewer suggestion, we performed SRIM analysis [13] of 16 MeV proton irradiation on 316L SS at the fluence of 20 μAh . The irradiation was performed at room temperature. The results are shown below in terms of displacement damage per atom (dpa) as function of tube thickness. The average dpa is around 0.04 dpa across the tube. This level of irradiation damage at room temperature does create point defects (interstitials and vacancies) in the materials but is too low to generate extended defects such as voids and large dislocation loops, as well as to generate extended microchemical segregation [14]. Indeed, in 316L SS, a proton irradiation dose of 1 to 1.5 dpa is considered close to the saturation of the dislocation loops and the radiation induced segregations phenomenon, where effects on post-corrosion rate can be observed [15]. In our study, the damage is about 25 times less than that. In addition, irradiation-induced point defects quickly anneal out (recombine or diffuse to sinks) at 620°C during the corrosion exposure. Consequently, that low level of radiation in the material is not expected to affect post-irradiation corrosion rate.

Figure: SRIM results of displacement per atom (dpa) as function of tube thickness. Note the total thickness of the tube in our study is 150 μ m and is indicated by the line labelled “tube thickness). A zoomed-in plot is provided on the right side.

The above description has been added into page 15 of the revised manuscript.

- **Comment 10:** Figure 1d – it is not clear to the reader at what positions the thickness of the thinned down tube section was measured. The choice for the location points can be parallel to the 1.5 cm long section or along the lateral tube surface – please specify in the text.
- ✓ **Response:** The positions where the thickness of the thinned down tube section was measured including locations being parallel to the 1.5 cm long section as well as along the lateral tube surface. This statement has been specified in the method section in page 15 of the revised manuscript.
- **Comment 11:** gamma spec needs a more detailed description. ICS-P4 is only a designation that the detector head came with an internal cooling system, i.e. was electrically cooled. There is no specific information about the Ge crystal given, such as relative efficiency, resolution etc. Please provide this information using the associated Ortec datasheets.
- ✓ **Response:** The HPGe detector used is a p-type aluminum windowed ORTEC GEM detector with a relative efficiency at 1333 keV of about 10% and a measured gamma peak FWHM at 1333 keV of 1.99 keV. This description has been added in page 17 of the revised manuscript.
- **Comment 12:** It is stated that the detector was calibrated for energy and efficiency using an Am-241 reference source. That is by far insufficient to be able to quantify the activity of radionuclides mentioned in this manuscript. Please consult a person proficient in gamma spectrometry and elaborate on the methodology these detector(s) were calibrated with.

✓ **Response:** The reviewer is correct and we thank them for pointing this error in the manuscript. The detector was actually calibrated for energy and efficiency using an Eu-152 reference source. Below, we've included additional data of calibration spectra and efficiency calibration values for one detector, showing the single point source detector energy and efficiency calibration was conducted using Eu-152. We greatly appreciate the reviewer pointing this out and have corrected it in page 17 of the revised manuscript accordingly.

Eu-152 Emission Data

Energy (keV)	Branching Ratio (%)
121.7817	28.53
244.6974	7.55
344.2785	26.59
411.1165	2.292
443.9606	3.152
778.9045	12.93
867.38	4.23
964.057	14.51
1085.837	10.11
1112.076	13.67
1299.142	1.633

IAEA. Live Chart of Nuclides: nuclear structure and decay data.

- **Comment 13:** Some further comments:
Line 130: If I is the beam current, given in [μA], then equation (1) should also contain the time t before the brackets. The product $I \cdot t$ in [$\text{A s} = \text{Coulomb}$] is the total charge transmitted to a target.
- ✓ **Response:** We thank the reviewer for pointing this out. We have edited the paragraph below the equation (1) with “ I is the total number of particles incident on the target in protons/s” in the revised manuscript.
- **Comment 14:** Line 135: The activity profiles in Figure 2a is given in $\mu\text{Ci}/\mu\text{Ah}$, not $\text{nCi}/\mu\text{Ah}$. consider using SI units Bq.
- ✓ **Response:** We thank the reviewer for pointing this out. The unit in Figure 2a has been replaced with the SI units Bq and corrected in the revised manuscript accordingly. At the same time, the manuscript has been checked out thoroughly to confirm the unit of μCi has been replaced with the SI unit Bq.
- **Comment 15:** Line 152: What is diffusion-induced radioisotope corrosion? Chemically and physically, there is no difference between radioactive and stable isotopes of elements in a vast majority of processes. Diffusion-induced corrosion will influence the erosion of stable Mn-55, present 2 wt% in 316L SS, the same way as it would for Mn-52. The radioactive isotopes in this manuscript are used as in-situ tracers, that behave exactly as the according element.
- ✓ **Response:** We totally agree with the reviewer and recognize that the wording is misleading. We were trying to express that the diffusion-induced corrosion (not radioisotope diffusion-

induced corrosion) into the salt is likely negligible. This wording has been revised accordingly throughout the revised manuscript.

- **Comment 16:** However, radiotracers decay – and the sentence in line 153 is somewhat misleading. The authors mention it later, but unless decay correction is applied, the activity loss observed experimentally will mainly come from decay loss (even for Co-56). Please rephrase this sentence and add decay correction.
- ✓ **Response:** All the activity data presented in the manuscript are reported after decay correction (see response to Comment 1 from reviewer 3). The misleading sentence has been rephrased as “Thus, the activity loss after decay correction observed experimentally should be mainly induced by surface recession rather than by diffusion-induced corrosion.” in page 5 of the revised manuscript.
- **Comment 17:** Line 175 – Figure 2a: Why there is a sudden change in the slope for all three curves at roughly 120 μm depth? Figure 2b: Why do you plot the concentration of Mn-52, and not stable Mn-55 (see also comment above)? It is not clear to the reviewer why it was important to prove the negligible effect from diffusion induced Mn loss using an additional figure. An explanation in the text could have been sufficient. Figure 2c – what is the form of the Bragg curve passing 150 μm of 316L SS? Would it be possible to indicate the mean proton energy exiting Foil 12?
- ✓ **Response:** The energy of the proton beam passing through the 316 L SS will decrease as a function of tube thickness. At the thickness of 120 μm , the energy of the 16 MeV proton beam is attenuated to about 14 MeV. Based on the TENDL tabulated cross sections as shown below, there is a significant change on the cross sections for ^{52}Mn and ^{51}Cr productions at 14 MeV. This results in the sharp changes in ^{52}Mn and ^{51}Cr activity profiles (also represent concentration profiles) at the thickness of 120 μm .

This explanation has been added in page 4 of the revised manuscript.

At high-temperatures, radionuclides would diffuse in the alloy following random-walk theory and mix with the stable ^{55}Mn . This would alter the activity (or concentration) profiles in Figure 2(a) and the relative activity profile in Figure 2(d) used to determine the corrosion attack depth. The negligible diffusion of ^{52}Mn shown in Figure 2(b) indicates that the relative activity profile in Figure 2(d) will not be affected by diffusion and initial concentration conditions at the end of bombardment can be utilized to determine the corrosion attack depth. This explanation has been added in page 5 of the revised manuscript.

Regarding the Bragg curve and associated mean proton energy exiting the foil, as mentioned above, we have performed SRIM analysis of 16 MeV protons through 316L SS (the proton fluence is not relevant to this comment). The Bragg peak is at about $550 \mu\text{m}$ (see the figure in the response to Comment 9), well beyond the $150 \mu\text{m}$ foil thickness. The distribution of proton energy exiting the last foil computed from SRIM is plotted below and the mean is at about 13.4 MeV.

Figure: SRIM results of transmitted proton energy distribution from a 16 MeV proton beam passing through 150 μm of 316L SS

- **Comment 18:** Line 183: It is stated that the experiment was running for 260 h, after which the cold leg temperature reached values below the eutectic melting point. It is a bit unclear if this was out of intention or it happened accidentally, stopping the whole endeavor prematurely. Could you indicate what was the true reason for the melt temperature to drop?
- ✓ **Response:** Due to the small size (1/4 inch) of the tube used for constructing the loop, the corrosion products are known to build up at the cold leg and eventually clog the tube with the loop operation. As the flow cross section drops, the local temperature drops, and eventually the salt freezes, resulting in the end of the flow corrosion test. This is not a premature stop; this is how most experiment stops in micro-loop studies. Similar phenomena were also observed in many other loops as reported by our collaborators in their study [16].

To make it clear, the previous sentence has been rephrased as “The molten salt microloop circulated naturally for approximately 260 hours with the hot leg maintained at 620 °C and the coldest section at around 500 °C until it succumbed to corrosion product buildup and clogging. Subsequently, the temperature in the cold leg decreased to values below the NaCl-MgCl₂ melting point, resulting in the loss of natural circulation.” in the revised manuscript.

- **Comment 19:** Line 192 downwards:
The authors make here conclusions on the type of the underlying Mn corrosion and the recession depth based on the observed decrease in measured, decay-corrected Mn-52 activity over time. It is not clear to the reviewer why – as common practice in scientific studies like this – an uncertainty estimation was not performed first before making any

claims towards interpretation of the observed phenomena. It is stated that a decrease of 2% is observed for the Mn-52 activity between start and end of the experiment. However, gamma spectrometry typically comes with an uncertainty of at least 3-5% for quantitative activity measurements, originating from the uncertainty of the calibration source activity, reported emission probability uncertainties, the counting statistics, etc. Even in relative measurements, where detector efficiency and emission probabilities cancel out, short, medium and long-term variations of temperature or humidity of the room or detector HV induce an uncertainty contribution that cannot be neglected. Please consult H.C. Lepy et al.: Uncertainties in gamma-ray spectrometry, Metrologia, Vol 52(3) 2015 for further details.

- ✓ **Response:** We thank the reviewer for these valuable comments and we agree that uncertainty considerations were lacking in the original manuscript. The primary source of data uncertainty in this study stems from the counting statistical calculation of the full energy peak area of the gamma-ray spectra for each radionuclide. In the revised manuscript, we have re-analyzed all the gamma-ray spectra data and incorporated data uncertainty using the method outlined below.

As shown below, to calculate the full energy peak area, the peak channel ± 3 FWHM was selected as the region of interest (ROI) [17].

The background area B is given by equation (1):

$$B = \left(\sum_{i=l}^{l+(n-1)} C_i + \sum_{i=h-(n-1)}^h C_i \right) \frac{h-l+1}{2n} \quad (1)$$

where l is the ROI low limit, h is the ROI high limit, C_i is the counts of channel i , n is the number of background points which is set to 5 in this study as suggested in the literature [17]. The adjusted gross area A_{ag} is the sum of all the channels marked by the ROI but not used in the background according to the equation (2) [17]

$$A_{ag} = \sum_{i=l+n}^{h-n} C_i \quad (2)$$

The net area A_n is the adjusted gross area minus the adjusted calculated background, as follows [17]

$$A_n = A_{ag} - \frac{B(h-l-(2n-1))}{(h-l+1)} \quad (3)$$

The uncertainty in the net area is the square root of the sum of the squares of the uncertainty in the adjusted gross area and the weighted error of the adjusted background. The background uncertainty is weighted by the ratio of the adjusted peak width to the number of channels used to calculate the adjusted background. Thus, net peak-area uncertainty σ_{A_n} is given by [17]:

$$\sigma_{A_n} = \sqrt{A_{ag} + B \left(\frac{h-l-(2n-1)}{2n} \right) \left(\frac{h-l-(2n-1)}{h-l+1} \right)} \quad (4)$$

The activity of each radionuclide was derived from the net area A_n and the associated uncertainty was derived from σ_{A_n} using error propagation.

The above method description and the re-analyzed activity for each radionuclide associated with the data uncertainty (shaded area), as shown below, have been added in the revised manuscript.

We agree with the reviewer that gamma spectrometry typically comes with an uncertainty of at least 3-5% for quantitative activity measurements, originating from the uncertainty of the calibration source activity, reported emission probability uncertainties, the counting statistics, etc. However, this 3-5% uncertainty is the error of the measured activity from the “actual” value of the activity. As noted by the reviewer, the present study focuses on the relative changes in activity, in which detector efficiency and emission probabilities uncertainty contributions cancel out. Short, medium and long-term variations of temperature or humidity of the room or detector high voltage HV can indeed induce uncertainty [18]. However, these uncertainty contributions should affect all the gamma-ray spectra relatively homogeneously rather than impacting one radionuclide peak in particular (^{52}Mn), but not the others (^{51}Cr and ^{56}Co). In addition, the corrosion experiment was performed in a shielded and closed basement laboratory area in which the temperature and humidity are quite stable during the whole, and relatively short testing period (see the recorded room temperature variation during the experiment period shown below). Moreover, the sealing of the semiconductor in the capsule of our HPGe detectors should help prevent the influence from humidity. The fixed HV value of our HPGe detector, set throughout the test, should also prevent uncertainty contribution from the detector HV. Therefore, the uncertainty contributions from temperature, humidity, and detector HV should be negligible in this study. The counting statistical uncertainty added in the revised

manuscript should improve the quality of this manuscript, and we thank the reviewer for pointing this out.

Room temperature variations during the experiment period.

- **Comment 20:** It would have been important to have installed a fixed check source attached to the experimental setup that would provide “reference” lines from a long-lived isotope, i.e. Co-60 or Eu-152 for each HPGe measurement to monitor its stability over time. It is obvious from Figure 1a, that especially in the last 40 h of the experiment there is a considerable variation of the measured, decay corrected Mn-52 activity. If this is due to counting statistics, an oscillation effect in the measurement over time or due to other effects, needs to be checked in more detail. However, the paper cannot be accepted as such due to the obvious neglect of the uncertainty contribution. Please rewrite this section and the resulting conclusions accordingly.
- ✓ **Response:** We are not sure what stability the reviewer is referring to specifically. HPGe detectors typically drift slightly with conditions in terms of energy calibration, but their efficiency in a well-controlled environment as ours should change very little, if at all. Since we are not dealing with interfering gamma lines, the former concern is easily accounted for by manual analysis (as opposed to automated or algorithmic peak detection). In addition, as noted in the previous response, considering that Co will be barely corroded in molten chloride salt because of its high thermodynamical stability, the decay-corrected activity variation of Co-56 can serve as a proxy for monitoring the detector’s stability over time. The Co-56 signal does vary over time, but this variation is clearly random over the 260 hours of exposure, while it is clear from visual inspection that the Mn-52 signal decreases over time. We do agree with the reviewer that it will be better to take a long-lived isotope

such as Co-60 or Eu-152 as a reference to monitor the detector's stability. This will be considered in our future work.

Due to the short half-life of Mn-52 ($t_{1/2} = 5.6$ d), the full energy peak of Mn-52 in the gamma spectra becomes significantly smaller with time, which would bring a higher error in counting statistics. This accounts for the variation of the measured, decay corrected Mn-52 activity in the last 40 h of the experiment.

Like our response to comment 19, the uncertainty contribution has been added and the corresponded description has been rewritten in the revised manuscript. We hope this revision could improve the quality of this manuscript.

- **Comment 21:** Some further comments:
Line 203: It appears that in Figure 3c, the activity for Cr-51 seems to rise – an effect that is likely explained considering different uncertainty contributions. It is unclear why here the authors state that it “remains stable”, while the apparent relative change is much more pronounced if compared to Figure 3a. It seems someone wants to trick the reader plotting the y-axis in 3a
- ✓ **Response:** As mentioned above, the activity of radionuclides has been re-analyzed and the uncertainty has been added. The scale of the axis was also revised for easy comparison and the activity variation of Cr-51 was found to be relatively stable as what we stated previously.
- **Comment 22:** Line 214: Reference [29] is a website that might likely disappear in the future. Please use a published book or a scientific paper, for example I. Barin: Thermochemical Data of Pure Substances, VCH, Berlin (1995).
- ✓ **Response:** The reference has been changed in the revised manuscript.
- **Comment 23:** Line 217: Figures 3a-d contain two distinct graphs each, however no explanation is given what is the difference. Avoid the usage of the unit [Ci], use the SI [Bq] instead. It is even recommended not to report activities in these graphs at all and rather refer to activity ratios relative to experiment start. What is the correlation factor between the full energy peak count rates of Mn-52 and Cr-51? There appears to be a dip in activity for both isotopes at 150 h, is this a coincidence?
- ✓ **Response:** We agree with the reviewer that reporting relative activity is more correct to compare the radionuclides together. Based on the obtained activity and the initial activity at the beginning of corrosion experiment, the relative activity has been calculated. The

calculated relative activity for each radionuclide is reported in the revised manuscript, while the activity is provided in the supplementary document for reader's reference.

We have given the correlation plot between the ^{52}Mn activity and ^{51}Cr activity as shown below, from which a weak correlation can be observed. The correlation factor was calculated as well and the value is about 0.039.

- **Comment 24:** Line 242 – Figure 4c: It would be beneficial to keep the color coding of the elements as in Figure 4a and plot on the x-axis point 1,2,3 and before and use the color for the elements. This should improve the readability of that data.
- ✓ **Response:** Based on the reviewer's comment, Figure 4c has been replaced with the new Figure below to keep the same color coding and improve readability.

- Comment 25:** Line 252: There is no full clarity how interference from the irradiated pipe section or other parts of the microloop was suppressed for activity determinations at P1, P2 and P3. From Figure 4S there appears to be some shielding at various positions at various thickness. Were there tests performed if the installed Pb shielding thickness is sufficient to sufficiently suppress gamma lines originating from Cr-51, Mn-52 or Co-56? It is likely much easier to do so for Cr-51 compared to the others. Would it be possible to measure the total activity of Mn-52 in the loop (P1+P2+P3) and determine the recession depth from the ratio to its initial activity at the welded section? An ISOCS characterized detector will likely be required for this purpose.
- ✓ **Response:** These are excellent comments and were also mentioned by reviewer #3. In our study, the thickness of the lead sheet shielding around the loop is 6.4 mm while that of the lead tube set on the HPGe detector is 3 cm. Based on the half value layer (the thickness of the material at which the intensity of the gamma-ray reduced by one half) of lead as a function of gamma-ray energy as shown below [9], the half value layer of lead for ^{51}Cr ($\gamma = 320 \text{ keV}$), ^{52}Mn ($\gamma = 936 \text{ keV}$), and ^{56}Co ($\gamma = 1037 \text{ keV}$) are about 0.18 cm, 0.8 cm, and 0.92 cm, respectively. The thickness of the lead sheet around the main radioactive source in the loop (the irradiated tube section), 6.4 mm, is about 3.6 times the half value layer of lead for ^{51}Cr . This means only about 8% of the ^{51}Cr gamma-rays are able to escape the irradiated tube lead shielding (to clarify, the gamma-ray detector for the irradiated tube has a direct field of view on the irradiated tube). The much thicker lead tube set on the HPGe detector serves as an additional collimator to only measure gamma-rays originating from the loop sections directly in front of the detector. In addition, the distance between the detector (for activity measurement at P1, P2, and P3) and the irradiated tube section can

also decrease the interference of gamma-rays emitting from the irradiated tube section. Therefore, the shielding design in this study is considered sufficient to shield gamma-rays originating from ^{51}Cr and the measured activity of ^{51}Cr at P1, P2 and P3 should be barely interfered by other loop sections.

Half value layer of lead as a function of gamma-ray energy [9].

Escape ratio of gamma-rays from different radionuclides under different shields.

Shielding thickness	escape ratio		
	^{51}Cr ($\gamma = 320$ keV)	^{52}Mn ($\gamma = 936$ keV)	^{56}Co ($\gamma = 1037$ keV)
6.4 mm	8%	57%	62%
3 cm	~0	7%	10%

On the other hand, for ^{52}Mn and ^{56}Co , about 57% and 62% of the original gamma rays would transmit through the lead sheet, respectively. Although the thickness of the lead tube collimator (3 cm) set on the detector can shield 93% of the gamma-rays from ^{52}Mn , and 90% of the gamma-rays from ^{56}Co , the detector can still accept gamma-rays from its front end. There is a large amount of gamma-rays being measured at P1, P2, and P3 originating from the irradiated tube. As shown in the Figure below, there are measurable differences in activity of ^{52}Mn and ^{60}Co at P1, P2 and P3, but the measurements are likely too convoluted to be confidently interpreted. Considering the high value of the activity of ^{52}Mn and ^{60}Co at P1, P2 and P3 as shown below, the measured activity should be mainly from the irradiated tube section. This also explained the close value of the activity of ^{52}Mn and ^{60}Co as shown below (the activity of ^{52}Mn and ^{60}Co at the irradiated tube section are close to each other). In the Figure below, the difference on the activity of ^{52}Mn and ^{56}Co at different locations should be resulted from the different distances between the detector and

the irradiated tube section and the different shielding angle of the lead tube relatively to the irradiated tube sections for these three different locations.

Figure: Data uncertainty is not visible due to the large y-axis scale. Note that the ^{52}Mn and ^{56}Co activity at all points are quite close from one another, and that they are also quite close from one another in the irradiated tube (see activity variations of these three radionuclides in the Supplementary Document), reinforcing the fact that these radioisotope activities at P1, P2, and P3 are convoluted with the activities from the irradiated tube.

The gamma spectra for P1, P2, and P3 were measured before the corrosion experiment and shown as below (gamma-ray spectra from P1 as an example) to verify the sufficiency of the lead shielding design in this study. In the Figure shown below, no ^{51}Cr full energy peak ($\gamma = 320 \text{ keV}$) was found, indicating the lead shielding is sufficient to avoid the interference of ^{51}Cr from the irradiated tube section. However, as what we discussed above, signals of ^{52}Mn and ^{56}Co resulted from the interference of the irradiated tube section were detected. The counts ratio of Mn and Co from the P1 and the irradiated tube section are even about 0.5 and 0.54, respectively. This high interference is the main reason why the transport of ^{52}Mn and ^{56}Co could not be studied in this work.

The solid salt taken from the microloop were analyzed after corrosion testing using an ISOCS characterized detector. The total activity of ^{52}Mn in the salt was measured to be about $1.85 \mu\text{Ci}$, which is close to the activity loss of ^{52}Mn at the irradiated tube section (about $2 \mu\text{Ci}$). This shows the feasibility of using this method to determine the recession depth. However, this method cannot be used *in-situ* and also assumes no redeposition of the specific radionuclide of interest on other parts of the loop during the test.

The manuscript has been revised accordingly based on the above discussion in page 10 of the revised manuscript.

- **Comment 26:** Line 256: The authors state that the Cr-51 signal is decreasing from P1 to P3, likely due to the decrease of Cr concentration within the salt. Is there any possibility to retrieve a sample of the salt to underline this statement? Cr transport is likely due to dissolution/precipitation in the eutectic melt at different positions of the loop, but *in-situ* gamma spectrometry is unable to determine if the origin of the Cr-51 signal is the salt, the solid-liquid interface or the 316L itself.
- ✓ **Response:** Based on the reviewer’s comment, we retrieved one salt sample from the tube section at P2 at the end of test, and measured the activity of ^{51}Cr using an ISOCS characterized HPGGe detector. The signal of ^{51}Cr in the salt was indeed detected (although very small) and its activity is about $0.2 \pm 0.004 \text{ nCi/g}$. We do agree with the reviewer that the *in situ* gamma spectrometry in this study is unable to determine if the origin of the Cr-51 signal is the salt or the solid-liquid interface or the 316L itself, which is also one weakness of this *in situ* method.

- **Comment 27:** Line 262: The authors state that one order of magnitude less (10%) of Cr-51 activity is within the detection noise of the HPGe. Are you sure about the absolute activities stated in Figure 5b? Anyhow, it is still confusing to read your conclusions drawn with Mn-52, where only 2% of loss was measured ...
- ✓ **Response:** The measured ^{51}Cr activity in the salt at P1, P2, and P3 is on the order of the random activity fluctuations of ^{51}Cr at the irradiated tube section (about 20,000 Bq, see Figure S2 in the supplementary document). This is likely the reason why the activity of ^{51}Cr was detected at P1, P2, and P3 while its activity loss was not statistically observed in the irradiated tube. The statement in page 10 of the revised manuscript should clarify this point.

Due to the relatively gentle corrosion attack, we agree with the reviewer that the activity loss of 2% of ^{52}Mn is too small and the detection uncertainty itself could reach to that value. This is indeed one weakness of the present study. However, the detection uncertainty might not bring a regular decline to the activity of ^{52}Mn as what we observed in this study. In addition, the present study focuses on the relative measurement and the influence of detection uncertainty will be much limited. Therefore, the activity loss of ^{52}Mn should be mainly resulted from the corrosion and this was also verified by the material characterizations on the post-corroded tube. Our response to comments 19 and 25 related to peak area uncertainty and data fluctuations also reinforced that the Mn-52 decreasing activity trend is meaningful relative to the random noise.

While not yet published, a longer exposure test with more severe corrosion attack is currently being performed by us to study the effect of different testing environments on the material corrosion in molten salt loop. While random noise is observed in the data as well, a significant, decay-corrected activity loss of ^{52}Mn is clearly observed as shown below. However, this new study is still preliminary and is beyond the research scope of method development in the present study. We invite the reviewer to follow our future work performed in different testing conditions using the method developed in this study.

The above discussion has been added into the revised manuscript accordingly to improve the quality of this manuscript.

- **Comment 28:** Line 267: Reference [34] applies to LBE corrosion, where other phenomena dominate. Comparing your observations to effects reported for liquid metal corrosion is like comparing apples and pears. Please look out for a reference describing molten salt corrosion.
- ✓ **Response:** The reference has been replaced with one describing molten salt corrosion (Jinsuo Zhang. "Systematic Corrosion Model for Non-isothermal Molten Salt Loop". TMS 2023 Annual Meeting & Exhibition, San Diego, March 19-23, 2023).
- **Comment 29:** Line 277 – Figure 5c: For someone not familiar with SEM images, it would be helpful to indicate where is the bulk and where the interface layer.
- ✓ **Response:** Based on the reviewer's comment, the interface layer and bulk region has been indicated in the revised manuscript.
- **Comment 30:** Line 312 downwards:

The authors give references to other works that compares corrosion depth to the (statistically questionable) stated recession depth of 3.5 μm . To the reviewers opinion, however, the most important outcome of the work is the confirmation of Cr dissolution and re-deposition at different sections of the loop using this in-situ technique. It is surprising why there was no further data provided for Co-56 and Mn-52, since these could have given further insights on the corrosion behavior of these two elements in molten salt loops.

- ✓ **Response:** We agree with the reviewer that there are two main conclusions in this work, one related to in-situ corrosion monitoring using Mn-52 in the irradiated tube, and another related to mass transport of Cr-51 in the loop. As mentioned in our response to comment 25, the activity of ^{56}Co and ^{52}Mn detected at P1, P2, and P3 are significantly convoluted with the activity from irradiated tube section. Therefore, the activity variation of ^{56}Co and ^{52}Mn for these three locations are not usable (data is still shared in the revised supplementary document for the reader's reference).
- **Comment 31:** Reactor irradiation of stainless steel samples would create the possibility to introduce samples of much higher activity and more homogeneous distribution for isotopes Cr-51 and Fe-59. Although this technique is associated with a number of challenges, the authors could provide more insights about potential benefits for future molten salt corrosion studies, giving examples of existing ones from nuclear power plants (see references above).

Response: We agree with the reviewer that using reactor neutron irradiation instead of accelerator proton irradiation to create a more homogeneous radioisotope profile would be an interesting path forward. As a first step in developing this new and challenging approach, we focused on accelerator proton irradiation to facilitate the workflow. Indeed, the workflow for this experiment is extremely challenging as it requires, (i) irradiation, (ii) cool-down, (iii) welding tube on the microloop, (iv) reassembling microloop insulation and thermocouples, (v) loading salt in glovebox, (vi) moving the loop from a rad-lab to a shielded room, (vii) start the loop, all in about 10 days to prevent significant decay of relevant radionuclides. Adding irradiation in a reactor was deemed too ambitious at this moment. However, this could now be envisioned and a sentence of "It should be noted that the radionuclides produced in this study are from proton irradiation. Neutron irradiation in reactor could create a more homogeneous radionuclide profile through the tube thickness, and this would be an interesting path moving forward to provide more insights to future corrosion studies." been added to the end of the discussion section.

- **Comment 32:** It is a stony way towards improvement, but many thanks for your hard work!

- ✓ **Response:** We thank again the effort from the reviewer to help us improve the quality of this manuscript. The comments were of great quality and certainly identified weak points within our data acquisition and analysis. We hope the reviewer agrees that our (sometimes lengthy) responses and revisions to the manuscript have strengthened the paper and made it worthy of publication.

All these discussions have been reflected and highlighted in yellow in revised manuscript. Again, we really appreciate the time and effort from editor and reviewers.

References

- [1] Ren, Sen, Yanjun Chen, Xiang-Xi Ye, Li Jiang, Shuai Yan, Jianping Liang, Xinmei Yang et al. "Corrosion behavior of carburized 316 stainless steel in molten chloride salts." *Solar Energy* 223 (2021): 1-10.
- [2] Shankar, A. Ravi, and U. Kamachi Mudali. "Corrosion of type 316L stainless steel in molten LiCl–KCl salt." *Materials and corrosion* 59, no. 11 (2008): 878-882.
- [3] Aung, Naing Naing, and Xingbo Liu. "High temperature electrochemical sensor for in situ monitoring of hot corrosion." *Corrosion science* 65 (2012): 1-4.
- [4] Guo, Shaoqiang, Weiqian Zhuo, Yafei Wang, and Jinsuo Zhang. "Europium induced alloy corrosion and cracking in molten chloride media for nuclear applications." *Corrosion Science* 163 (2020): 108279.
- [5] Weisberg, Arel, Rollin E. Lakis, Michael F. Simpson, Leo Horowitz, and Joseph Craparo. "Measuring lanthanide concentrations in molten salt using laser-induced breakdown spectroscopy (LIBS)." *Applied spectroscopy* 68, no. 9 (2014): 937-948
- [6] Sridharan, Kumar, Mark Anderson, Michael Corradini, Todd Allen, Luke Olson, James Ambrosek, and Daniel Ludwig. Molten salt heat transport loop: materials corrosion and heat transfer phenomena. No. DOE/ID/14675. University of Wisconsin system, Madison, 2008.
- [7] Koning, Arjan, Rochman, Dimitri A., Sublet, J. Ch, Dzysiuk, Nataliya R., Fleming, Michael J., and van der Mark, Steven C. "TENDL: Complete Nuclear Data Library for Innovative Nuclear Science and Technology." *Nucl. Data Sheets*, no. 155 (2019) 1-55.
- [8] Aniĉin, I. V., and C. T. Yap. "New approach to detection limit determination in spectroscopy." *Nuclear Instruments and Methods in Physics Research Section A: Accelerators, Spectrometers, Detectors and Associated Equipment* 259, no. 3 (1987): 525-528.
- [9] Tekin, Hüseyin Ozan, Ghada ALMisned, Shams AM Issa, and Hesham MH Zakaly. "A rapid and direct method for half value layer calculations for nuclear safety studies using MCNPX Monte Carlo code." *Nuclear Engineering and Technology* 54, no. 9 (2022): 3317-3323.
- [10] Raiman, Stephen S., J. Matthew Kurley, Dino Sulejmanovic, Adam Willoughby, Scott Nelson, Keyou Mao, Chad M. Parish, M. Scott Greenwood, and Bruce A. Pint. "Corrosion of 316H stainless steel in flowing FLiNaK salt." *Journal of Nuclear Materials* 561 (2022): 153551.
- [11] Jose, M. T., T. Ravi, D. N. Krishnakumar, and V. Meenakshisundaram. "Radioactive contamination measurements of the primary sodium pipes in FBTR by gamma spectrometry." *Annals of Nuclear Energy* 36, no. 5 (2009): 641-649.
- [12] Gregorich, Carola. "Critical review of gamma spectrometry detection approaches for in-plant surface deposition monitoring." In 2015 4th International Conference on Advancements in Nuclear Instrumentation Measurement Methods and their Applications (ANIMMA), pp. 1-8. IEEE, 2015.

- [13] Stoller, Roger E., Mychailo B. Toloczko, Gary S. Was, Alicia G. Certain, Shyam Dwaraknath, and Frank A. Garner. "On the use of SRIM for computing radiation damage exposure." *Nuclear instruments and methods in physics research section B: beam interactions with materials and atoms* 310 (2013): 75-80.
- [14] F.A. Garner, 4.02 - Radiation Damage in Austenitic Steels, Editor(s): Rudy J.M. Konings, *Comprehensive Nuclear Materials*, Elsevier, 2012, Pages 33-95.
- [15] Boisson, Marylou, Laurent Legras, Eric Andrieu, and Lydia Laffont. "Role of irradiation and irradiation defects on the oxidation first stages of a 316L austenitic stainless steel." *Corrosion Science* 161 (2019): 108194.
- [16] Kelleher, Brian C., Sean F. Gagnon, and Ivan G. Mitchell. "Thermal gradient mass transport corrosion in NaCl-MgCl₂ and MgCl₂-NaCl-KCl molten salts." *Materials Today Communications* 33 (2022): 104358.
- [17] Gamma Vision. "Gamma-ray Spectrum Analysis and MCA Emulation for MS Windows 7, 8.1, and 10 Professional, A66-BW, A66SV-BW, A66MP-BW." *Software User's Manual*, Software version 9.
- [18] Uncertainties in gamma-ray spectrometry, *Metrologia*, Vol 52(3) 2015

REVIEWER COMMENTS

Reviewer #1 (Remarks to the Author):

My comments have been satisfactorily addressed. Nice work!

Reviewer #2 (Remarks to the Author):

In my first review I have highly estimated the work, and I am keeping my opinion on the manuscript. All the changes that authors have made have sufficiently improved the understanding of their investigations.

Reviewer #3 (Remarks to the Author):

I would like to extend my gratitude to the authors for their diligent efforts in addressing the review comments. The manuscript has undergone significant improvements. As highlighted by the authors, the application of in-situ gamma spectrometry to the micro high-temperature molten salt loop system presents a potentially novel approach. However, I would appreciate further clarification on whether the duration of the loop running time is sufficient to observe corrosion induced by thermally gradient. I believe that it is calculated by the activity profile change by thermally driven diffusion or diffusion-induced corrosion into the salt. In comparing this micro loop study to traditional loop work, it would be valuable to explore key factors such as temperature, operation time, the quantity of corrosion samples, and other relevant parameters. To gain a clearer understanding of corrosion progress, it is essential to convert activity results to concentrations. While it is unfortunate that this conversion is not feasible in the current work, I am curious about whether there are enough corrosion products accumulated to consume impurities and transition into states of corrosion driven by thermal gradients. Alternatively, do the authors suggest that thermally driven diffusion is negligible within a loop system?

Despite maintaining reservations about the specific new information revealed by this measurement, I acknowledge the significance of encouraging creative approaches within the research community. I believe that this work has the potential to make a meaningful contribution to the advancement of the field.

Reviewer #4 (Remarks to the Author):

Dear authors,

i was looking forward to your replies to the questions the reviewers have raised concerning your manuscript. You have put substantial effort in revising it, did a re-evaluation of the experimental data and have now strengthened your claims in the paper. Additionally, you have pointed out weaknesses of this in-situ technique, which is honorable. A good paper is not about how great your method is, but where the issues and pitfalls are, so others can learn from it and improve.

Most importantly, the data you have presented on the loss of ^{52}Mn over time and the concluded recession depth has now sufficient statistical weight to be considered as real. Especially the re-evaluated ^{56}Co data shows clearly a stable behaviour over time, as would be expected for this element in molten salt corrosion. You have included an uncertainty estimation, have corrected several mistakes in the text and included additional information for clarity.

I congratulate you for the excellent work and i believe the manuscript can now be accepted for publication, considering only minor changes that i would like to highlight.

At first, please use prefixes to the unit [Bq] in order to reduce the number of digits for stated activities, i.e. 20 kBq and not 20.000 Bq.

Second - you have stated the activities of ^{52}Mn , ^{56}Co and ^{51}Cr , as requested by reviewer 3, but the stated uncertainties are far too low. As you rightfully write, error propagation needs to be applied to take into account the uncertainty from counting statistics, the emission probabilities, the uncertainty of the ^{152}Eu activity etc. Typically, calibration sources are provided with 1-2% uncertainty - for ^{52}Mn and ^{56}Co you state to be below 0.5%, which does not seem right. Please also consider indicating maximum 1 or 2 digits for the absolute uncertainty value, for example 2.8 ± 0.1 MBq or 2.8(1) MBq (= concise notation).

Third comment concerns your stated LoD values. In gamma spectrometry, the level of nuclide detection depend on the emission probability of each line of that nuclide. For ^{56}Co you have used the 1038 keV line with 14% emission probability, but it also emits a line at 847 keV with 99.9% probability - using that one would reduce your ^{56}Co LoD at least by a factor 7. Please indicate the energy of the line used for each stated LoD.

Apart from that, all my comments and questions from the previous review have been addressed and answered. I thank the authors for the invitation to follow their work and wish them lot of success in the future.

Authors' Responses to the Review Comments

Manuscript NCOMMS-23-47066A

Title: Radionuclide Tracing Based in situ Corrosion and Mass Transport Monitoring of 316L Stainless Steel in a Molten Salt Closed Loop

Authors: Yafei Wang, Aeli P. Olson, Cody Falconer, Brian Kelleher, Ivan Mitchell, Hongliang Zhang, Kumar Sridharan, Jonathan W. Engle, Adrien Couet

Feb 17, 2024

We thank the editor and the reviewers for their time, effort, and valuable comments. The manuscript has been revised based on the reviewers' comments and the revised part has been highlighted in yellow. We hope that these revisions improve the paper such that you deem it worthy of publication in Nature Communications.

Response to the reviewers' comments

Reviewer #1:

- ***Summary Comment:*** My comments have been satisfactorily addressed. Nice work!
- ✓ Response: We thank the reviewer very much and are delighted that the revised manuscript has addressed the reviewer's concerns.

Reviewer #2:

- ***Summary Comment:*** In my first review I have highly estimated the work, and I am keeping my opinion on the manuscript. All the changes that authors have made have sufficiently improved the understanding of their investigations
- ✓ Response: We thank the reviewer very much for the positive comment and sincerely appreciate the great support on the publication of our manuscript.

Reviewer #3:

- ***Summary Comment:*** I would like to extend my gratitude to the authors for their diligent efforts in addressing the review comments. The manuscript has undergone significant improvements. As highlighted by the authors, the application of in-situ gamma spectrometry to the micro high-temperature molten salt loop system presents a potentially novel approach. However, I would appreciate further clarification on whether the duration of the loop running time is sufficient to observe corrosion induced by thermally gradient. I

believe that it is calculated by the activity profile change by thermally driven diffusion or diffusion-induced corrosion into the salt. In comparing this micro loop study to traditional loop work, it would be valuable to explore key factors such as temperature, operation time, the quantity of corrosion samples, and other relevant parameters. To gain a clearer understanding of corrosion progress, it is essential to convert activity results to concentrations. While it is unfortunate that this conversion is not feasible in the current work, I am curious about whether there are enough corrosion products accumulated to consume impurities and transition into states of corrosion driven by thermal gradients. Alternatively, do the authors suggest that thermally driven diffusion is negligible within a loop system?

Despite maintaining reservations about the specific new information revealed by this measurement, I acknowledge the significance of encouraging creative approaches within the research community. I believe that this work has the potential to make a meaningful contribution to the advancement of the field.

- ✓ **Response:** We thank the reviewer for the recognition of the work we did for the improvement of this manuscript. In response to this additional comment, we think that the 260 hours of exposure is enough to activate the thermal gradient driven corrosion in the loop. The main evidence of this is the deposition of corrosion products that is observed at the cold leg and the negligible corrosion at the medium temperature sections (top and bottom sections of the loop) as shown below, which is similar as the phenomenon commonly observed in the traditional thermal gradient driven molten salt corrosion loop [1,2,3]. That deposition is resulted from the corrosion product activities in the salt at the cold leg being too high and moving the equilibrium toward the product metallic state and thus its deposition on the cold leg wall. Thus, a “fresh” salt re-enters the hot leg. The low activity of corrosion products in the salt at high temperature when the salt re-enters the hot leg drives further corrosion of the tube elements into the salt. If impurities would drive the corrosion, the salt redox potential would be quite high and redeposition would be less likely.

The further clarification has been added in page 6 of the revised manuscript as “The operation time of 260 hours is sufficient to activate the thermal gradient driven corrosion in the loop based on the material characterizations of different tube sections along the loop: deposition of corrosion products at cold leg, severe corrosion attack at hot leg, and negligible corrosion at the medium temperature sections (top and bottom sections of the loop) as observed in Figure S2 in the supplementary document and discussed in detailed later.”

We thank reviewer #3 again for carefully checking our manuscript and hope that the revisions improve the paper such that the reviewer deems it worthy of publication.

Reviewer #4:

- **Summary Comment:** I was looking forward to your replies to the questions the reviewers have raised concerning your manuscript. You have put substantial effort in revising it, did a re-evaluation of the experimental data and have now strengthened your claims in the paper. Additionally, you have pointed out weaknesses of this in-situ technique, which is honorable. A good paper is not about how great your method is, but where the issues and pitfalls are, so others can learn from it and improve.

Most importantly, the data you have presented on the loss of ^{52}Mn over time and the concluded recession depth has now sufficient statistical weight to be considered as real. Especially the re-evaluated ^{56}Co data shows clearly a stable behavior over time, as would be expected for this element in molten salt corrosion. You have included an uncertainty estimation, have corrected several mistakes in the text and included additional information for clarity.

I congratulate you for the excellent work and I believe the manuscript can now be accepted for publication, considering only minor changes that I would like to highlight.

- ✓ **Response:** We thank the reviewer very much for the valuable suggestions that have greatly helped us to improve the manuscript. We are very happy to see that the reviewer agrees with the publication of this manuscript. Based on the minor revision change suggestions from the reviewer, the manuscript was revised accordingly as below. We hope that these revisions could further improve the paper.
- **Comment 1:** At first, please use prefixes to the unit [Bq] in order to reduce the number of digits for stated activities, i.e. 20 kBq and not 20.000 Bq.
- ✓ **Response:** The unit of Bq has been replaced with kBq in the revised manuscript and highlighted in yellow in page 7 of the revised manuscript.
- **Comment 2:** Second - you have stated the activities of ^{52}Mn , ^{56}Co and ^{51}Cr , as requested by reviewer 3, but the stated uncertainties are far too low. As you rightfully write, error propagation needs to be applied to take into account the uncertainty from counting statistics, the emission probabilities, the uncertainty of the ^{152}Eu activity etc. Typically, calibration sources are provided with 1-2% uncertainty - for ^{52}Mn and ^{56}Co you state to be below 0.5%, which does not seem right. Please also consider indicating maximum 1 or 2 digits for the absolute uncertainty value, for example 2.8 ± 0.1 MBq or 2.8(1) MBq (= concise notation).
- ✓ **Response:** We thank the reviewer for this valuable comment. As what we answered to the reviewer in the previous response letter, this study only considered the data uncertainty stemming from the counting statistical calculation of the full energy peak area of the gamma-ray spectra since relative measurement was in the focus of the present work. This is the main reason why the stated uncertainties are far too low. The calculation of relative activity is based on the decay-corrected initial activity. All the uncertainties of activity data in the corrosion process reported in this study are only sourced from the counting statistical calculation. Therefore, we believe it might be better to keep the data uncertainty sources same for all the data.

However, to make it clearer, we have added a sentence of “data uncertainty is only sourced from counting statistical calculation of the full energy peak area of gamma-ray spectra” in page 7 of the revised manuscript. At the same time, unit of Bq has been replaced with kBq as the reviewer stated in comment 1.

- **Comment 3:** Third comment concerns your stated LoD values. In gamma spectrometry, the level of nuclide detection depend on the emission probability of each line of that nuclide. For ^{56}Co you have used the 1038 keV line with 14% emission probability, but it also emits a line at 847 keV with 99.9% probability - using that one would reduce your ^{56}Co LoD at least by a factor 7. Please indicate the energy of the line used for each stated LoD.

- ✓ **Response:** We thank the reviewer for this comment. The energy line used for each stated LoD has been added in page 7 of the revised manuscript.
- **Comment 4:** Apart from that, all my comments and questions from the previous review have been addressed and answered. I thank the authors for the invitation to follow their work and wish them lot of success in the future.
- ✓ **Response:** We thank the reviewer for the efforts to improve our manuscript and are very happy to see all the concerns from the reviewer has been satisfactorily addressed.

We thank again for the reviewer #4's valuable comments to improve the quality of this manuscript and the recommendation of the publication of our paper.

All the changes have been reflected and highlighted in yellow in revised manuscript. Again, we really appreciate the time and effort from editor and reviewers.

References

- [1] Sridharan, K., and T. R. Allen. "Corrosion in molten salts." In *Molten salts chemistry*, pp. 241-267. Elsevier, 2013.
- [2] Guo, Shaoqiang, Jinsuo Zhang, Wei Wu, and Wentao Zhou. "Corrosion in the molten fluoride and chloride salts and materials development for nuclear applications." *Progress in Materials Science* 97 (2018): 448-487.
- [3] Raiman, Stephen, J. Matthew Kurley, Dino Sulejmanovic, Adam Willoughby, Scott Nelson, Keyou Mao, Chad Parish, M. Scott Greenwood, and Bruce Pint. "Corrosion of 316H Stainless Steel in Flowing FLiNaK Salt." (2021).

REVIEWERS' COMMENTS

Reviewer #3 (Remarks to the Author):

All of my concerns are properly addressed. I would like to thank the authors for their diligent efforts.

Reviewer #4 (Remarks to the Author):

Dear authors,

many thanks for your quick response. You have partly taken my remarks into account. Note, however, that a reader proficient in gamma spectrometry will ask himself why only the uncertainty of the total peak area was considered in the uncertainty budget of the stated activities. Since you have explicitly stated your approach it is fine, however keep in mind that it is still considered to be an incomplete uncertainty budget.

After consistently unifying the usage of the prefix "k" (you use kBq but somewhere else KJ/mol) i advised the editor to proceed with the publication of your manuscript. Once again, i thank you for your work and congratulate you for this manuscript.

Authors' Responses to the Review Comments

Manuscript NCOMMS-23-47066B

Title: Radionuclide Tracing Based in situ Corrosion and Mass Transport Monitoring of 316L Stainless Steel in a Molten Salt Closed Loop

Authors: Yafei Wang, Aeli P. Olson, Cody Falconer, Brian Kelleher, Ivan Mitchell, Hongliang Zhang, Kumar Sridharan, Jonathan W. Engle, Adrien Couet

March 5, 2024

We thank the editor and the reviewers for their time, effort, and valuable comments. We are so glad all the concerns from the reviewers have been addressed and this manuscript can be in principle published in Nature Communications.

Response to the reviewers' comments

Reviewer #3:

- **Summary Comment:** All of my concerns are properly addressed. I would like to thank the authors for their diligent efforts.
- ✓ Response: We thank you very much and are delighted that our revised manuscript has addressed all your concerns.

Reviewer #4:

- **Summary Comment:** Many thanks for your quick response. You have partly taken my remarks into account. Note, however, that a reader proficient in gamma spectrometry will ask himself why only the uncertainty of the total peak area was considered in the uncertainty budget of the stated activities. Since you have explicitly stated your approach it is fine, however keep in mind that it is still considered to be an incomplete uncertainty budget. After consistently unifying the usage of the prefix "k" (you use kBq but somewhere else KJ/mol) i advised the editor to proceed with the publication of your manuscript. Once again, i thank you for your work and congratulate you for this manuscript.
- ✓ Response: We thank the reviewer very much for the valuable suggestions and efforts to improve our manuscript. The manuscript has been checked through carefully and the prefix "k" has been unified. Thank you again for your agreement on the publication of our manuscript.